# Fast and accurate annotation of acoustic signals with deep neural networks

**Elsa Steinfath[1,2], Adrian Palacios-Muñoz[1,2], Julian R Rottschäfer[1,2], Deniz Yuezak[1,2], Jan Clemens[1,3]***

[1]European Neuroscience Institute - A Joint Initiative of the University Medical Center Göttingen and the Max-Planck-Society, Göttingen, Germany; [2]International Max Planck Research School and Göttingen Graduate School for Neurosciences, Biophysics, and Molecular Biosciences (GGNB) at the University of Göttingen, Göttingen, Germany; [3]Bernstein Center for Computational Neuroscience, Göttingen, Germany

**Abstract** Acoustic signals serve communication within and across species throughout the animal kingdom. Studying the genetics, evolution, and neurobiology of acoustic communication requires annotating acoustic signals: segmenting and identifying individual acoustic elements like syllables or sound pulses. To be useful, annotations need to be accurate, robust to noise, and fast. We here introduce *DeepAudioSegmenter* (*DAS*), a method that annotates acoustic signals across species based on a deep-learning derived hierarchical presentation of sound. We demonstrate the accuracy, robustness, and speed of *DAS* using acoustic signals with diverse characteristics from insects, birds, and mammals. *DAS* comes with a graphical user interface for annotating song, training the network, and for generating and proofreading annotations. The method can be trained to annotate signals from new species with little manual annotation and can be combined with unsupervised methods to discover novel signal types. *DAS* annotates song with high throughput and low latency for experimental interventions in realtime. Overall, *DAS* is a universal, versatile, and accessible tool for annotating acoustic communication signals.

***For correspondence:**
clemensjan@gmail.com

**Competing interests:** The authors declare that no competing interests exist.

## Introduction

Animals produce sounds to foster group cohesion (*Haack et al., 1983*; *Janik and Slater, 1998*; *Chaverri et al., 2013*), to signal the presence of food, friend, or foe (*Cäsar et al., 2013*; *Clay et al., 2012*), and to find and evaluate mating partners (*Baker et al., 2019*; *Behr and von Helversen, 2004*; *Holy and Guo, 2005*; *Sangiamo et al., 2020*). Studying acoustic communication not only provides insight into social interactions within and across species; it can also reveal the mechanisms driving complex behaviors: The genetics and evolution of signal production and recognition (*Ding et al., 2016*), the genes and circuits driving song learning (*Kollmorgen et al., 2020*), or the fast and precise sensorimotor transformations involved in vocal interactions (*Coen et al., 2016*; *Cator et al., 2009*; *Fortune et al., 2011*; *Okobi et al., 2019*). The first step in many studies of acoustic communication is song annotation: the segmentation and labeling of individual elements in a recording. Acoustic signals are diverse and range from the repetitive long-distance calling songs of crickets, grasshoppers, and anurans (*Gerhardt and Huber, 2002*), the dynamic and context-specific courtship songs of vinegar flies or rodents (*Coen et al., 2014*; *Clemens et al., 2018*; *Neunuebel et al., 2015*; *Sangiamo et al., 2020*), to the complex vocalizations produced by some birds and primates (*Lipkind et al., 2013*; *Weiss et al., 2014*; *Landman et al., 2020*).

This diversity in signal structure has spawned a zoo of annotation tools (*Arthur et al., 2013*; *Coffey et al., 2019*; *Tachibana et al., 2020*; *Goffinet et al., 2021*; *Koumura and Okanoya, 2016*; *Cohen et al., 2020*), but existing methods still face challenges: First, assessing vocal repertoires and

their relation to behavioral and neural dynamics (*Clemens et al., 2018*; *Coffey et al., 2019*; *Neunuebel et al., 2015*; *Fortune et al., 2011*; *Okobi et al., 2019*) requires annotations to be complete and temporally precise even at low signal levels, but annotation can fail when signals are weak (*Coffey et al., 2019*; *Stern et al., 2017*). Second, analyses of large datasets and experimental interventions during behavior (*Fortune et al., 2011*; *Okobi et al., 2019*; *Bath et al., 2014*; *Tschida and Mooney, 2012*; *Stowers et al., 2017*) need annotations to be fast, but existing methods are often slow. Last, annotation methods should be flexible and adaptable (*Ding et al., 2016*; *Ding et al., 2019*; *Clemens et al., 2018*; *Clemens and Hennig, 2013*), but existing methods often work only for restricted types of signals or adapting them to new signals requires tedious manual tuning (*Clemens et al., 2018*).

In brief, an accurate, fast, and flexible framework for annotating song across species is missing. A general framework would not only improve upon existing methods but would also facilitate the study of species for which automated methods do not yet exist. Deep neural networks have emerged as powerful and flexible tools for solving data annotation tasks relevant for neuroscience such as object recognition, pose tracking, or speech recognition (*Krizhevsky et al., 2012*; *Graves and Jaitly, 2014*; *Mathis et al., 2018*; *Pereira et al., 2019*; *Graving et al., 2019*). These methods are not only fast and accurate but also easily adapted to novel signals by non-experts since they only require annotated examples for learning. Recently, deep neural networks have also been used for annotating animal vocalizations (*Oikarinen et al., 2019*; *Coffey et al., 2019*; *Cohen et al., 2020*; *Sainburg et al., 2020*; *Arthur et al., 2021*; *Goffinet et al., 2021*).

We here present a new deep-learning-based framework for annotating acoustic signals, called *Deep Audio Segmenter* (*DAS*). We test the framework on a diverse set of recordings from insects, birds, and mammals, and show that *DAS* annotates song in single- and multi-channel recordings with high accuracy. The framework produces annotations with low latency on standard PCs and is therefore ideally suited for closed-loop applications. Small-to-moderate amounts of manual annotations suffice for adapting the method to a new species and annotation work can be simplified by combining *DAS* with unsupervised methods. We provide *DAS* as an open-source software package with a graphical user interface for manually annotating audio, training the network, and inferring and proofreading annotations. Integration into existing frameworks for signal analysis or experimental control is possible using a programmatic interface. The code and documentation for *DAS* are available at https://janclemenslab.org/das/.

## Results

### Architecture and working principle of *DAS*

Acoustic signals are defined by features on multiple timescales—the fast harmonic oscillations of the sound carrier (<10 ms), modulations of amplitude (AM) and frequency (FM) (10–100 ms), and the sequencing of different AM and FM patterns into bouts, syllables, or phrases (10–1000 ms). These patterns are typically made explicit using a hand-tuned pre-processing step based on time-resolved Fourier or wavelet transforms (*Arthur et al., 2013*; *Van Segbroeck et al., 2017*; *Coffey et al., 2019*; *Oikarinen et al., 2019*; *Cohen et al., 2020*). Most deep-learning-based methods then treat this pre-defined spectrogram as an image and use methods derived from computer vision to extract the AM and FM features relevant for annotation (*Oikarinen et al., 2019*; *Coffey et al., 2019*; *Cohen et al., 2020*). Recurrent units are sometimes used to track the sound features over time (*Cohen et al., 2020*). This approach can produce accurate annotations but has drawbacks: First, the spectrogram constitutes a strong and proven pre-processing step, but it is unsuitable for some signal types, like short pulsatile signals. Second, the pre-processing transform is typically tuned by hand and may therefore require expert knowledge for it to produce optimal results. Lastly, the recurrent units used in some methods (*Cohen et al., 2020*) excel at combining information over time to provide the context information necessary to annotate spectrally complex signals, but they can be hard to train and slow to run (*Bai et al., 2018*).

*DAS* solves these limitations in three ways: First, the pre-processing step is optional. This makes *DAS* more flexible, since signals for which a time-resolved Fourier transform is not appropriate—for instance, short pulsatile signals—can now also be processed. Second, the optional preprocessing step is integrated and optimized with the rest of the network. This removes the need to hand-tune

this step and allows the network to learn a preprocessing that deviates from a time-resolved Fourier or wavelet transform if beneficial (*Choi et al., 2017*). Integrating the preprocessing into the network also increases inference speed due to the efficient implementation and hardware acceleration of deep-learning frameworks. Third and last, *DAS* learns a task-specific representation of sound features using *temporal convolutional networks* (TCNs) (*Bai et al., 2018*; *van den Oord et al., 2016*; *Guirguis et al., 2021*; *Figure 1—figure supplement 1A–E*). At the core of TCNs are so-called *dilated convolutions* (*Yu and Koltun, 2016*). In standard convolutions, short templates slide over the signal and return the similarity with the signal at every time point. In *dilated* convolutions, these templates have gaps, allowing to analyze features on longer timescales without requiring more parameters to specify the template. Stacking dilated convolutions with growing gap sizes results in a hierarchical, multi-scale representation of sound features, which is ideally suited for the hierarchical and harmonic structure of animal vocalizations.

The output of the deep neural network in *DAS* is a set of confidence scores for each audio sample, corresponding to the probability of each song type (*Figure 1C*). Annotation labels for the different song types are mutually exclusive and are produced by comparing the confidence score to a threshold or by choosing the most probable song type. Brief gaps in the annotations are closed and short spurious detections are removed to smoothen the annotation. For song types that are described as events, like the pulses in fly song (*Figure 1A*), the event times are extracted as local maxima that exceed a confidence threshold.

## *DAS* accurately annotates song from a diverse range of species
### Fly courtship song
We first tested *DAS* on the courtship song of *Drosophila melanogaster*, which consists of two major modes (*Figure 1A*): The sine song, which corresponds to sustained oscillations with a species-specific carrier frequency (150 Hz), and two types of pulse song, which consists of trains of short (5–10 ms) pulses with carrier frequencies between 180 and 500 Hz, produced with a species-specific interval (35–45 ms in *D. melanogaster*). Males dynamically choose the song modes based on sensory feedback from the female (*Coen et al., 2014*; *Clemens et al., 2018*; *Calhoun et al., 2019*). Despite the relative simplicity of the individual song elements, an accurate annotation of fly song is challenging because of low signal-to-noise ratio (SNR): The song attenuates rapidly with distance (*Bennet-Clark, 1998*) and is highly directional (*Morley et al., 2018*), which can lead to weak signals if the male is far from the microphone (*Figure 1A*). Moreover, the interactions between the flies introduce pulsatile noise and complicate the accurate and complete annotation of the pulse song.

We first trained *DAS* to detect the pulse and the sine song recorded using a single microphone (data from *Stern, 2014*) and compared the performance of *DAS* to that of the current state-of-the-art in fly song segmentation, *FlySongSegmenter* (*FSS*) (*Arthur et al., 2013*; *Coen et al., 2014*; *Clemens et al., 2018*). Annotation performance was quantified using *precision*, the fraction of correct detections, and *recall*, the fraction of true song that is detected (*Figure 1E,J*, *Figure 1—figure supplement 1F,G*). We counted detected pulses within 10 ms of a true pulse as correct detections. Ten ms corresponds to 1/4th of the typical interval between pulses in a train and results are robust to the choice of this value (*Figure 1—figure supplement 2A*). *DAS* detects pulses with a high precision of 97% - only 3% of all detected pulses are false detections - and a high recall of 96% - it misses only 4% of all pulses. This is a substantial improvement in recall over *FSS*, which has slightly higher precision (99%) but misses 13% of all pulses (87% recall) (*Figure 1D,E*). In *DAS*, the balance between precision and recall can be controlled via the confidence threshold, which corresponds to the minimal confidence required for labeling a pulse (*Figure 1C*): Lowering this threshold from 0.7 to 0.5 yields a recall of 99% for pulse song with a modest reduction in precision to 95%. The performance gain of *DAS* over *FSS* for pulse stems from better recall at high frequencies (>400 Hz) and low SNR (*Figure 1G,H*). To assess *DAS* performance for sine song, we evaluated the sample-wise precision and recall. *DAS* has similar precision to *FSS* (92% vs 91%) but higher recall (98% vs. 91%) (*Figure 1I, J*). Recall is higher in particular for short sine songs (<100 ms) and at low SNR (<1.0) (*Figure 1L,M*). The performance boost for pulse and sine arises because *DAS* exploits context information, similar to how humans annotate song: For instance, *DAS* discriminates soft song pulses from pulsatile noise based on the pulse shape but also because song pulses occur in regular trains while noise pulses do not (*Figure 1—figure supplement 2C*). A comparison of *DAS*' performance to that of human

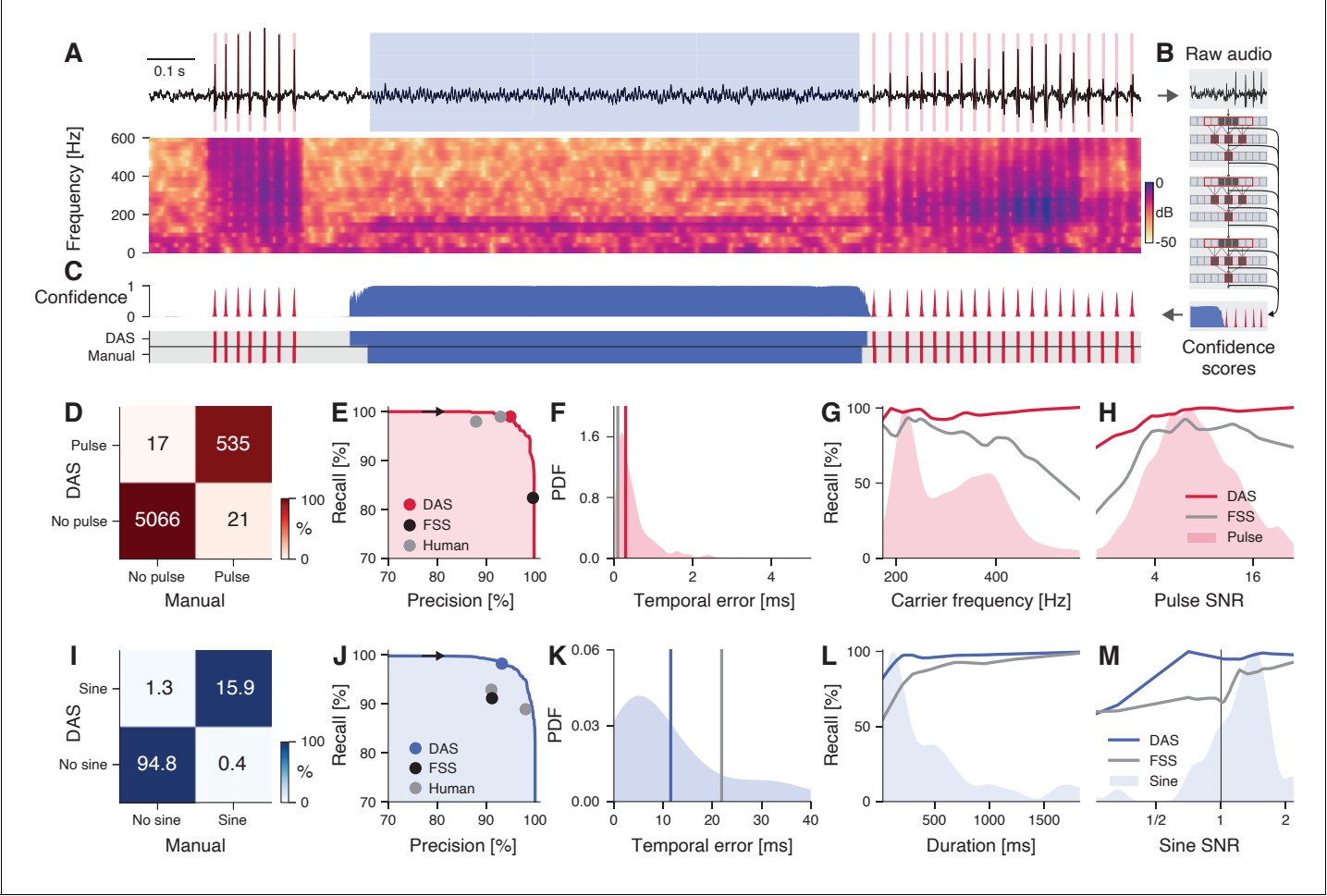

**Figure 1.** DAS performance for fly song. (A) Fly song (black, top) with manual annotations of sine (blue) and pulse (red) song. The spectrogram (bottom) shows the signal's frequency content over time (see color bar). (B) *DAS* builds a hierarchical presentation of song features relevant for annotation using a deep neural network. The network consists of three TCN blocks, which extract song features at multiple timescales. The output of the network is a confidence score for each sample and song type. (C) Confidence scores (top) for sine (blue) and pulse (red) for the signal in A. The confidence is transformed into annotation labels (bottom) based on a confidence threshold (0.5 for sine, 0.7 for pulse). Ground truth (bottom) from manual annotations shown for comparison. (D) Confusion matrix for pulse from the test data set. Color indicates the percentage (see color bar) and text labels indicate the number of pulses for each quadrant. All confusion matrices are normalized such that columns sum to 100%. The concentration of values along the diagonal indicates high annotation performance. (E) Precision-recall curve for pulse depicts the performance characteristics of *DAS* for different confidence thresholds (from 0 to 1, black arrow points in the direction of increasing threshold). Recall decreases and precision increases with the threshold. The closer the curve to the upper and right border, the better. The red circle corresponds to the performance of *DAS* for a threshold of 0.7. The black circle depicts the performance of FlySongSegmenter (*FSS*) and gray circles the performance of two human annotators. (F) Probability density function of temporal errors for all detected pulses (red shaded area), computed as the distance between each pulse annotated by *DAS* and the nearest manually annotated pulse. Lines depict the median temporal error for *DAS* (red line, 0.3 ms) and *FSS* (gray line, 0.1 ms). (G, H) Recall of *DAS* (red line) and *FSS* (gray line) as a function of the pulse carrier frequency (G) and signal-to-noise ratio (SNR) (H). Red shaded areas show the distributions of carrier frequencies (G) and SNRs (H) for all pulses. *DAS* outperforms *FSS* for all carrier frequencies and SNRs. (I) Same as in D but for sine. Color indicates the percentage (see color bar) and text labels indicate seconds of sine for each quadrant. (J) Same as in E but for sine. The blue circle depicts the performance for the confidence threshold of 0.5. (K) Distribution of temporal errors for all detected sine on- and offsets. Median temporal error is 12 ms for *DAS* (blue line) and 22 ms for *FSS* (gray line). (L, M) Recall for *DAS* (blue line) and *FSS* (gray line) as a function of sine duration (L) and SNR (M). Blue-shaded areas show the distributions of durations and SNRs for all sine songs. *DAS* outperforms *FSS* for all durations and SNRs.

The online version of this article includes the following figure supplement(s) for figure 1:

**Figure supplement 1.** *DAS* architecture and evaluation.

**Figure supplement 2.** Performance and the role of context for annotating fly pulse song.

**Figure supplement 3.** Performance for multi-channel recordings of fly courtship song.

annotators reveals that our methods exceeds human-level performance for pulse and sine (*Figure 1E,J*, *Table 1*).

Temporally precise annotations are crucial, for instance when mapping sensorimotor transformations based on the timing of behavioral or neuronal responses relative to individual song elements (*Coen et al., 2014*; *Srivastava et al., 2017*; *Long and Fee, 2008*; *Benichov and Vallentin, 2020*). We therefore quantified the temporal error of the annotations produced by *DAS*. For pulse song, the temporal error was taken as the distance of each pulse annotated by *DAS* to the nearest true pulse. The median temporal error for pulse is 0.3 ms which is negligible compared to the average duration of a pulse (5–10 ms) or of a pulse interval (35–45 ms) (*Deutsch et al., 2019*). For sine song, the median temporal error for on- and offsets was 12 ms, which is almost half of that of *FSS* (22 ms). Sine song can have low SNR (*Figure 1M*) and fades in and out, making the precise identification of sine song boundaries difficult even for experienced manual annotators (see *Figure 1A,C*).

Recording song during naturalistic interactions in large behavioral chambers often requires multiple microphones (*Coen et al., 2014*; *Neunuebel et al., 2015*). To demonstrate that *DAS* can process multi-channel audio, we trained *DAS* to annotate recordings from a chamber tiled with nine microphones (*Coen et al., 2014*; *Figure 1—figure supplement 3*, *Figure 1—figure supplement 1B*). *DAS* processes multi-channel audio by using filters that take into account information from all channels simultaneously. As is the case for existing methods (*Arthur et al., 2013*), we achieved maximal performance by training separate networks for the pulse and for the sine song (*Table 2*). In multi-channel recordings, *DAS* annotates pulse song with 98% precision and 94% recall, and sine song with 97% precision and 93% recall, and matches the performance of *FSS* (*FSS* pulse precision/recall 99/92%, sine 95/93%) (*Figure 1—figure supplement 3D–L*). Annotations of multi-channel audio have high temporal precision for pulse (*DAS* 0.3 ms, *FSS* 0.1 ms) and sine (*DAS* 8 ms, *FSS* 15 ms) (*Figure 1—figure supplement 3E,J*). Overall, *DAS* performs better or as well as the current state-of-the-art method for annotating single and multi-channel recordings of fly song.

## Mouse ultrasonic vocalizations

Mice produce ultrasonic vocalizations (USVs) in diverse social contexts ranging from courtship to aggression (*Sangiamo et al., 2020*; *Warren et al., 2020*; *Neunuebel et al., 2015*). We tested *DAS* using audio from an intruder assay, in which an anesthetized female was put into the home cage and the USVs produced by a resident female or male were recorded (*Ivanenko et al., 2020*). The female USVs from this assay typically consist of pure tones with weak harmonics and smooth frequency modulations that are often interrupted by frequency steps (*Figure 2A,B*). The male USVs are similar but also contain complex frequency modulations not produced by the females in this assay (*Figure 2C,D*). Recording noise from animal movement and interaction as well as the frequency steps often challenge spectral threshold-based annotation methods and tend to produce false positive syllables (*Tachibana et al., 2020*; *Coffey et al., 2019*). Moreover, weak signals often lead to missed syllables or imprecisely delimited syllables. We first trained and tested *DAS* on recordings of a female mouse interacting with an anesthetized female intruder (*Figure 2A*). *DAS* annotates the female USVs with excellent precision (98%) and recall (99%) (*Figure 2E*) and low median temporal error (0.3 ms) (*Figure 2F*). *DAS* is robust to noise: Even for weak signals (SNR 1/16) the recall is 90% (*Figure 2G*). These performance values are on par with that of methods specialized to annotate USVs (*Tachibana et al., 2020*; *Coffey et al., 2019*; *Van Segbroeck et al., 2017*) (see *Table 3*). USVs of female and male residents have similar characteristics (*Figure 2A,B*) and the female-trained *DAS* network also accurately annotated the male vocalizations (*Figure 2H*). Notably, even the male

**Table 1.** Comparison to human annotators for fly song.
See also *Figure 1E,J*.

| Annotator | Sine recall [%] | Sine precision [%] | Pulse recall [%] | Pulse precision [%] |
|---|---|---|---|---|
| Human A | 89 | 98 | 99 | 93 |
| Human B | 93 | 91 | 98 | 88 |
| *FSS* | 91 | 91 | 87 | 99 |
| *DAS* | 98 | 92 | 96 | 97 |

**Table 2.** Precision, recall, and temporal error of *DAS*.
Precision and recall values are sample-wise for all except fly pulse song, for which it is event-wise. The number of classes includes the 'no song' class. (p) Pulse, (s) Sine.

| Species | Trained | Classes | Threshold | Precision [%] | Recall [%] | Temporal error [ms] |
|---|---|---|---|---|---|---|
| Fly single channel | Pulse (p) and sine (s) | 3 | 0.7 | 97/92 (p/s) | 96/98 (p/s) | 0.3/12 (p/s) |
| Fly multi channel | Pulse (p) | 2 | 0.5 | 98 | 94 | 0.3 |
| Fly multi channel | Sine (s) | 2 | 0.5 | 97 | 93 | 8 |
| Mouse | Female | 2 | 0.5 | 98 | 99 | 0.3 |
| Marmoset | five male-female pairs | 5 | 0.5 | 85 | 91 | 4.4 |
| Bengalese finch | four males | 49 (38 in test set) | 0.5 | 97 | 97 | 0.3 |
| Zebra finch | one males | 7 | 0.5 | 98 | 97 | 1.2 |

syllables with characteristics not seen in females in the paradigm were detected (*Figure 2D*). Overall, *DAS* accurately and robustly annotates mouse USVs and generalizes across sexes.

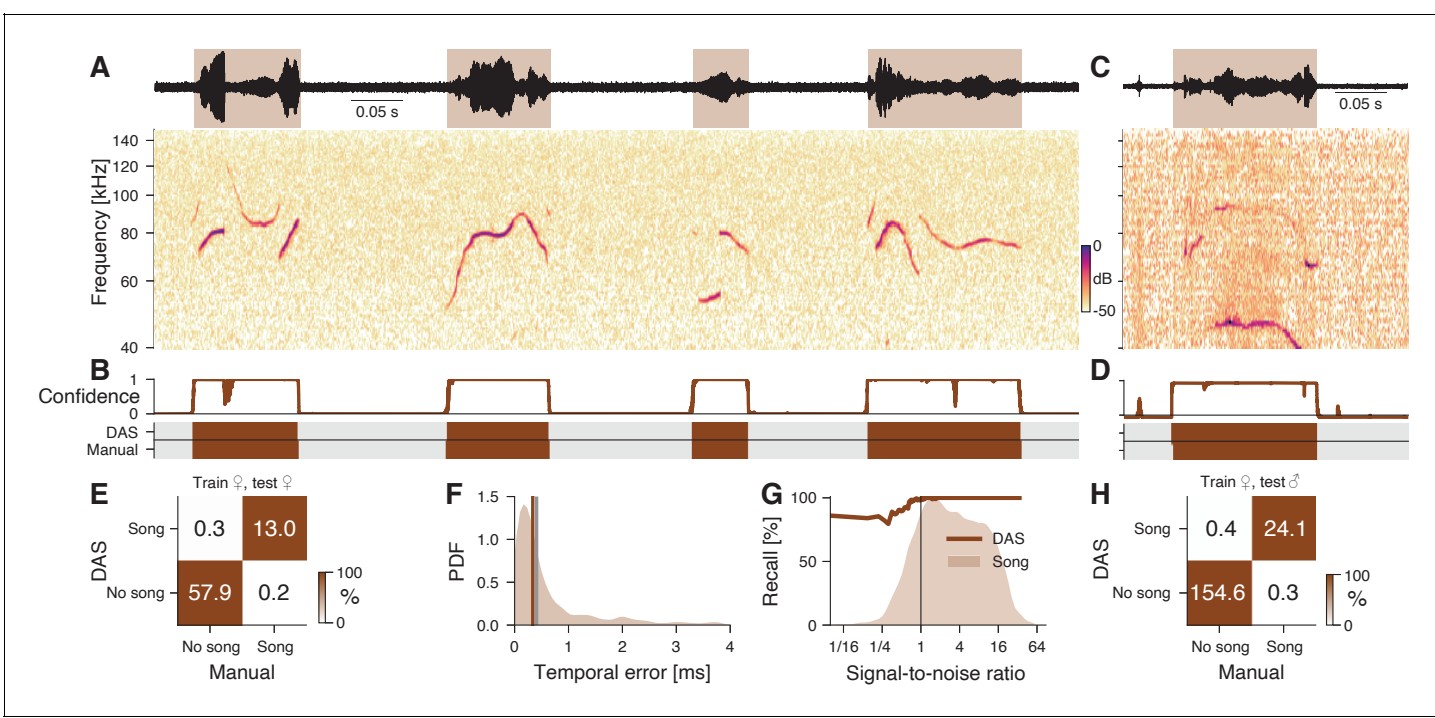

**Figure 2.** DAS performance for mouse ultrasonic vocalizations. (A) Waveform (top) and spectrogram (bottom) of USVs produced by a female mouse in response to an anesthetized female intruder. Shaded areas (top) show manual annotations. (B) Confidence scores (top) and *DAS* and manual annotations (bottom) for the female USVs in A. Brief gaps in confidence are filled smooth annotations. (C) Example of male USVs with sex-specific characteristics produced in the same assay. (D) Confidence scores (top) and *DAS* and manual annotations (bottom) for the male USVs in C from a *DAS* network trained to detect female USVs. (E) Confusion matrix from a female-trained network for a test set of female USVs. Color indicates the percentage (see color bar) and text labels the seconds of song in each quadrant. (F) Distribution of temporal errors for syllable on- and offsets in female USVs. The median temporal error is 0.3 ms for *DAS* (brown line) and 0.4 ms for *USVSEG Tachibana et al., 2020*, a method developed to annotate mouse USVs (gray line). (G) Recall of the female-trained network (brown line) as a function of SNR. The brown shaded area represents the distribution of SNRs for all samples containing USVs. Recall is high even at low SNR. (H) Confusion matrix of the female-trained *DAS* network for a test set of male USVs (see C, D for examples). Color indicates the percentage (see color bar) and text labels the seconds of song in each quadrant.

The online version of this article includes the following figure supplement(s) for figure 2:

**Figure supplement 1.** Performance for marmoset vocalizations.

**Table 3.** Comparison to alternative methods.

Methods used for comparisons: (1) *Arthur et al., 2013*, (2) *Tachibana et al., 2020*, (3) *Oikarinen et al., 2019*, (4) *Cohen et al., 2020*. (A,B) *DAS* was trained on 1825/15970 syllables which contained 4/7 of the call types from *Oikarinen et al., 2019*. (B) The method by *Oikarinen et al., 2019* produces an annotation every 50 ms of the recording - since the on/offset can occur anywhere within the 50 ms, the expected error of the method by *Oikarinen et al., 2019* is at least 12.5 ms. (C) The method by *Oikarinen et al., 2019* annotates 60 minutes of recordings in 8 minutes. (D) Throughput assessed on the CPU, since the methods by *Arthur et al., 2013* and *Tachibana et al., 2020* do not run on a GPU. (E) Throughput assessed on the GPU. The methods by *Cohen et al., 2020* and *Oikarinen et al., 2019* use a GPU.

| Species | Precision [%] | | Recall [%] | | Jitter [ms] | | Throughput [s/s] | |
|---|---|---|---|---|---|---|---|---|
| | *DAS* | Other | *DAS* | Other | *DAS* | Other | *DAS* | Other |
| Fly single (1) | 97/92 (p/s) | 99/91 | 96/98 (p/s) | 87/91 | 0.3/12 (p/s) | 0.1/22 | 15 | 4 (D) |
| Fly multi (1) | 98 | 99 | 94 | 92 | 0.3 | 0.1 | 8 (p) | 0.4 (p+s) (D) |
| Fly multi (1) | 97 | 95 | 93 | 93 | 8.0 | 15.0 | 8 (s) | 0.4 (p+s) (D) |
| Mouse (2) | 98 | 98 | 99 | 99 | 0.3 | 0.4 | 12 | 4 (D) |
| Marmoset (3) | 96 | 85 (A) | 92 | 77 (A) | 4.4 | 12.5 (B) | 82 | 7.5 (C, E) |
| Bengalese finch (4) | 99 | 99 | 99 | 99 | 0.3 | 1.1 | 15 | 5 (E) |
| Zebra finch (4) | 100 | 100 | 100 | 100 | 1.3 | 2.0 | 18 | 5 (E) |

## Marmoset vocalizations

We next examined the robustness of annotations produced by *DAS* to noisy recordings and variable vocalization types, by training a network to annotate vocalization from pairs of marmosets (*Landman et al., 2020*). The recordings contain lots of background noises like faint calls from nearby animals, overdrive from very loud calls of the recorded animals, and large variability within syllable types (*Figure 2—figure supplement 1A–D*). Recently, a deep-learning-based method was shown to produce good performance (recall 77%, precision 85%, 12.5 ms temporal error) when trained on 16,000 syllables to recognize seven vocalization types (*Oikarinen et al., 2019*). We trained *DAS* on 1/9th of the data (1800 syllables) containing four of the seven vocalization types. Despite the noisy and variable vocalizations, *DAS* achieves high syllable-wise precision and recall (96%, 92%, (*Figure 2—figure supplement 1E,F*)). Note that *DAS* obtains this higher performance at millisecond resolution (temporal error 4.4 ms, *Figure 2—figure supplement 1G*), while the method by *Oikarinen et al., 2019* only produces annotations with a resolution of 50 ms (*Table 2*).

## Bird song

Bird song is highly diverse and can consist of large, individual-specific repertoires. The spectral complexity and large diversity of the song complicates the annotation of syllable types. Traditionally, syllable types are annotated based on statistics derived from the segmented syllable spectrogram. Recently, good annotation performance has been achieved with unsupervised methods (*Sainburg et al., 2020*; *Goffinet et al., 2021*) and deep neural networks (*Koumura and Okanoya, 2016*; *Cohen et al., 2020*). We first trained *DAS* to annotate the song from four male Bengalese finches (data and annotations from *Nicholson et al., 2017*). The network was then tested on a random subset of the recordings from all four individuals which contained 37 of the 48 syllable types from the training set (*Figure 3A,B*, *Figure 3—figure supplement 1A,B*). *DAS* annotates the bird song with high accuracy: Sample-wise precision and recall are 97% and syllable on- and offsets are detected with sub-millisecond precision (median temporal error 0.3 ms, *Figure 3C*). The types of 98.5% the syllables are correctly annotated, with only 0.3% false positives (noise annotated as a syllable), 0.2% false negatives (syllables annotated as noise), and 1% type confusions (*Figure 3D*, *Figure 3—figure supplement 1C–D*). This results in a low sequence error (corresponding to the minimal number of substitutions, deletions, or insertions required to transform the true sequence of syllables into the inferred one) of 0.012. Overall, *DAS* performs as well as specialized deep-learning-based methods for annotating bird song (*Koumura and Okanoya, 2016*; *Cohen et al., 2020*, *Table 2*).

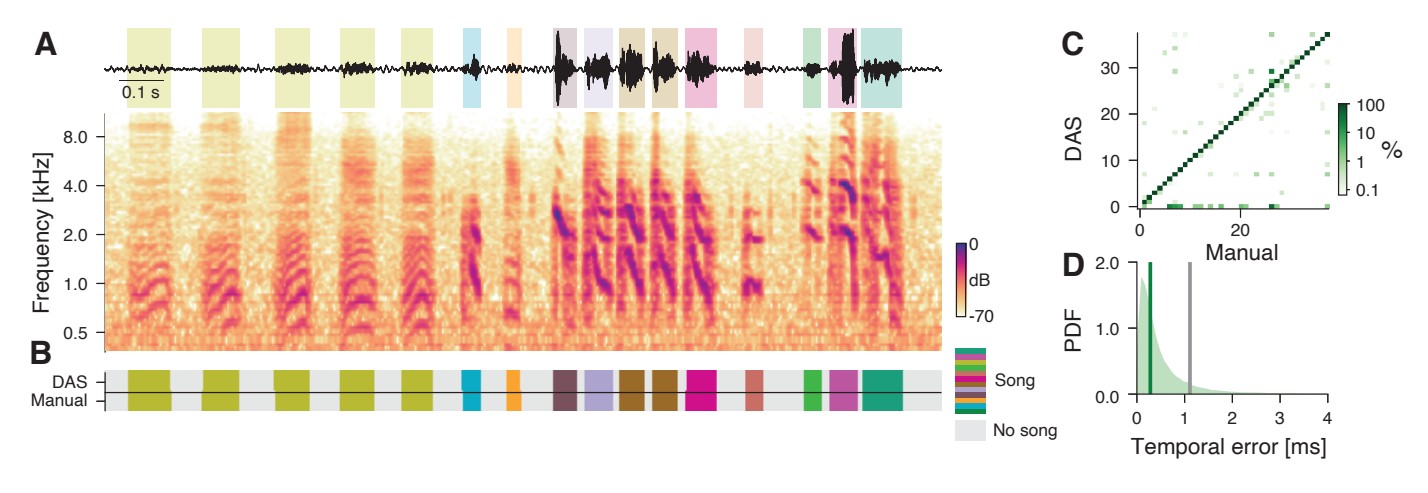

**Figure 3.** DAS performance for the song of Bengalese finches. (A) Waveform (top) and spectrogram (bottom) of the song of a male Bengalese finch. Shaded areas (top) show manual annotations colored by syllable type. (B) *DAS* and manual annotation labels for the different syllable types in the recording in A (see color bar). *DAS* accurately annotates the syllable boundaries and types. (C) Confusion matrix for the different syllables in the test set. Color was log-scaled to make the rare annotation errors more apparent (see color bar). Rows depict the probability with which *DAS* annotated each syllable as any of the 37 types in the test dataset. The type of 98.5% of the syllables were correctly annotated, resulting in the concentration of probability mass along the main diagonal. (D) Distribution of temporal errors for the on- and offsets of all detected syllables (green-shaded area). The median temporal error is 0.3 ms for *DAS* (green line) and 1.1 ms for TweetyNet *Cohen et al., 2020*, a method developed to annotate bird song (gray line).

The online version of this article includes the following figure supplement(s) for figure 3:

**Figure supplement 1.** Performance for the song of Bengalese finches.

**Figure supplement 2.** Performance for the song of a Zebra finch.

To further demonstrate the robustness of *DAS*, we trained a network to annotate song from Zebra finches. In Zebra finch males, individual renditions of a given syllable type tend to be more variable (*Fitch et al., 2002*). Moreover, the particular recordings used here (*Goffinet et al., 2021*) contain background noise from the bird's movement. Despite the variability and noise, *DAS* annotates the six syllables from a male's main motif with excellent precision and recall, and low temporal error, demonstrating that *DAS* is robust to song variability and recording noise (*Figure 3—figure supplement 2*).

In summary, *DAS* accurately and robustly annotates a wide range of signals—from the pulsatile song pulses of flies to the spectrally complex syllables of mammals and birds. *DAS* therefore constitutes a universal method for annotating acoustic signals that is as good as or better than methods specialized for particular types of signals (*Table 2*).

## *DAS* is fast

To efficiently process large corpora of recordings and to be suitable for closed-loop applications, *DAS* needs to infer annotations quickly. We therefore assessed the throughput and latency of *DAS*. Throughput measures the rate at which *DAS* annotates song and high throughput means that large datasets are processed quickly. Across the five species tested here, *DAS* has a throughput of 8-82x realtime on a CPU and of 24-267x on a GPU (*Figure 4A*). This means that a 60 min recording is annotated in less than five minutes on a standard desktop PC and in less than 1.5 minutes using a GPU, making the annotation of large datasets feasible (*Figure 4—figure supplement 1A–H*). The differences in throughput arise from the different sample rates and network architectures: The marmoset network is fastest because of a relatively shallow architecture with only 2 TCN blocks and a low sampling rate (44.1 kHz). By contrast, the multi-channel *Drosophila* networks have the lowest throughput because of multi-channel inputs (9 channels at 10.0 kHz) and a comparatively deep architecture with four TCN blocks (*Table 4*).

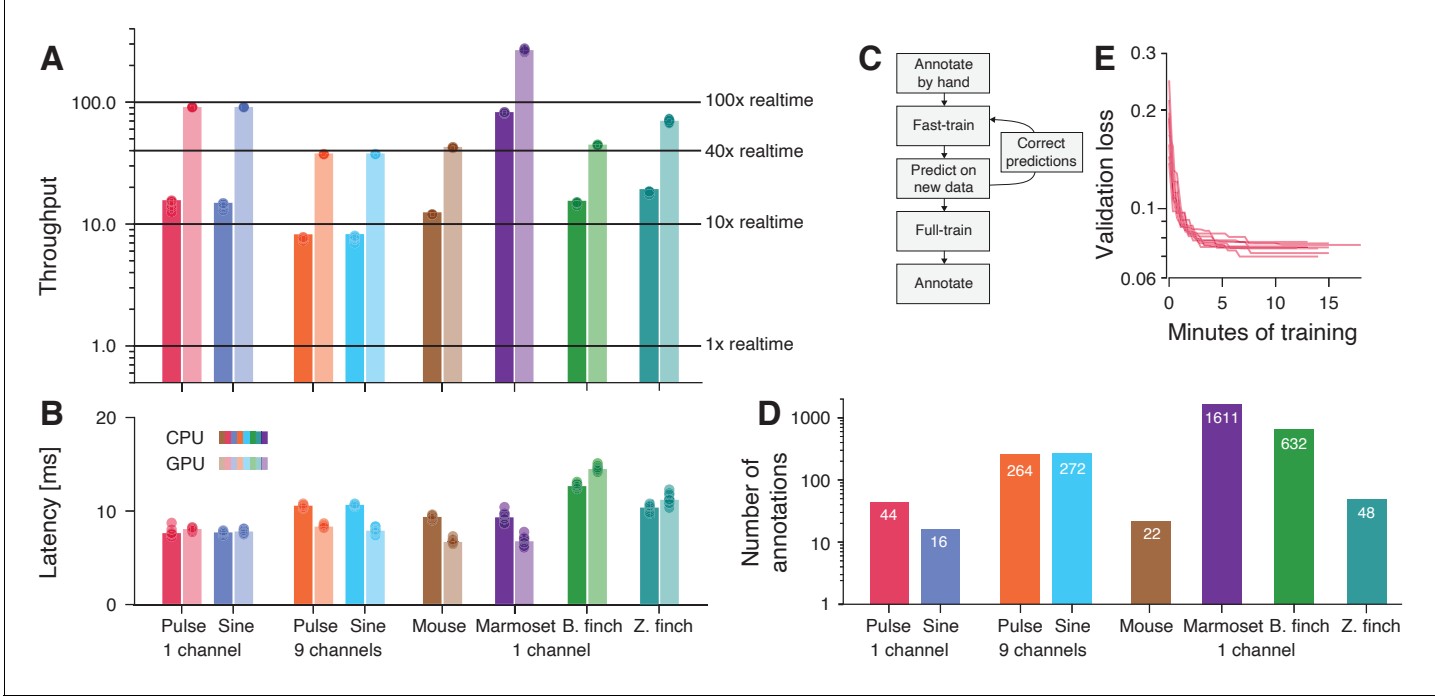

**Figure 4.** DAS annotates song with high throughput and low latency and requires little data. (**A, B**) Throughput (**A**) and latency (**B**) of *DAS* (see also *Figure 4—figure supplement 1*). Throughput (**A**) was quantified as the amount of audio data in seconds annotated in one second of computation time. Horizontal lines in A indicate throughputs of 1, 10, 40, and 100. Throughput is >8x realtime on a CPU (dark shades) and >24x or more on a GPU (light shades). Latency (**B**) corresponds to the time it takes to annotate a single chunk of audio and is similar on a CPU (dark shades) and a GPU (light shades). Multi-channel audio from flies was processed using separate networks for pulse and sine. For estimating latency of fly song annotations, we used networks with 25 ms chunks, not the 410 ms chunks used in the original network (see *Figure 1—figure supplement 2*). (**C**) Flow diagram of the iterative protocol for fast training *DAS*. (**D**) Number of manual annotations required to reach 90% of the performance of *DAS* trained on the full data set shown in *Figure 1*, *Figure 1—figure supplement 3*, *Figure 2*, *Figure 2—figure supplement 1*, *Figure 3*, and *Figure 3—figure supplement 2* (see also *Figure 4—figure supplement 3*). Performance was calculated as the F1 score, the geometric mean of precision and recall. For most tasks, *DAS* requires small to modest amounts of manual annotations. (**E**) Current best validation loss during training for fly pulse song recorded on a single channel for 10 different training runs (red lines, 18 min of training data). The network robustly converges to solutions with low loss after fewer than 15 min of training (40 epochs).

The online version of this article includes the following figure supplement(s) for figure 4:

**Figure supplement 1.** Throughput and latency of inference.
**Figure supplement 2.** Reducing the chunk duration reduces latency and comes with minimal performance penalties.
**Figure supplement 3.** *DAS* requires small to moderate amounts of data for training.
**Figure supplement 4.** Example of fast training mouse USVs.
**Figure supplement 5.** *DAS* performance is robust to changes in the structural parameters of the network.

**Table 4.** Structural parameters of the tested networks.

| Species | Rate [kHz] | Chunk [samples] | Channels | STFT downsample | Separable conv. | TCN stacks | Kernel size [samples] | Kernel |
|---|---|---|---|---|---|---|---|---|
| Fly single channel | 10.0 | 4096 | 1 | - | - | 3 | 32 | 32 |
| Fly multi channel (pulse) | 10.0 | 2048 | 9 | - | TCN blocks 1+2 | 4 | 32 | 32 |
| Fly multi channel (sine) | 10.0 | 2048 | 9 | - | TCN blocks 1+2 | 4 | 32 | 32 |
| Mouse | 300.0 | 8192 | 1 | 16x | - | 2 | 16 | 32 |
| Marmoset | 44.1 | 8192 | 1 | 16x | - | 2 | 16 | 32 |
| Bengales finch | 32.0 | 1024 | 1 | 16x | - | 4 | 32 | 64 |
| Zebra finch | 32.0 | 2048 | 1 | 16x | - | 4 | 32 | 64 |

A measure of speed crucial for closed-loop experiments is latency, which quantifies the time it takes to annotate a chunk of song and determines the delay for experimental feedback. Latencies are short, between 7 and 15 ms (*Figure 4B*, *Figure 4—figure supplement 1I–P*) on CPUs and GPUs. One network parameter impacting latency is the chunk size—the duration of audio processed at once—and we find that for fly song, latency can be optimized by reducing chunk size with a minimal impact on accuracy (*Figure 4—figure supplement 2*). The low latency of annotation makes *DAS* well suited for triggering realtime optogenetic or acoustic feedback upon the detection of specific vocalizations (*Bath et al., 2014*; *Stowers et al., 2017*).

We also compared the speed of *DAS* to that of other methods that were specifically developed to annotate the types of signals tested here. Since most existing methods are not suitable for estimating latency due to constraints in their design and interface, we only compared throughput. We find that *DAS* achieves 3x to 10x higher throughput than existing methods (*Table 2*). This has three main reasons: First, the relatively simple, purely convolutional architecture exploits the parallel processing capabilities of modern CPUs and GPUs. Second, Fourier or wavelet-like preprocessing steps are integrated into the network and profit from a fast implementation and hardware acceleration. Third, for multi-channel data, *DAS* combines information from all audio channels early, which increases throughput by reducing the data bandwidth.

Overall, *DAS* annotates audio with high throughput (>8x realtime) and low latency (<15 ms) and is faster than the alternative methods tested here. The high speed renders *DAS* suitable for annotating large corpora and for realtime applications without requiring specialized hardware.

## *DAS* requires little manual annotation

To be practical, *DAS* should achieve high performance with little manual annotation effort. We find that *DAS* can be efficiently trained using an iterative protocol (*Pereira et al., 2019*, *Figure 4C*, *Figure 4—figure supplement 4*): Annotate a small set of recordings and train the network for a few epochs; then generate annotations on a larger set of recordings and correct these annotations. Repeat the predict-correct-train cycle on ever larger datasets until performance is satisfactory. To estimate the amount of manual annotations required to achieve high performance, we evaluated *DAS* trained on subsets of the full training data sets used above (*Figure 4—figure supplement 3*). We then took the number of manual annotations needed to reach 90% of the performance of *DAS* trained on the full data sets as an upper bound on the data requirements (*Figure 4D*). With a performance threshold of 90%, the resulting networks will produce sufficiently accurate annotations for creating a larger body of training data with few corrections. Performance was taken as the F1 score, the geometric mean of precision and recall. For single-channel recordings of fly song, fewer than 50 pulses and 20 sine songs are needed to reach 90% of the performance achieved with the full data set. For mouse vocalizations, *DAS* achieves 90% of its peak performance with fewer than 25 manually annotated syllables. Even for the six syllables from a zebra finch, *DAS* reaches the 90% threshold with only 48 manually annotated syllables (eight per type). Manually annotating such small amounts of song for flies, mice, or zebra finches takes less than 5 min. Likewise, for the song of Bengalese finches, 1–51 (median 8, mean 17) manual annotations are required per syllable type, with one outlier requiring 200 syllables (*Figure 4—figure supplement 3C*). Closer inspection reveals that the outlier results from an annotation error and consists of a mixture of three distinct syllable types (*Figure 4—figure supplement 3D–F*). Even with this outlier, only 626 manually annotated syllabes (424 without) are required in total to reach 90% of the test performance of a network trained on >3000 annotated syllables. Data requirements are higher for the multi-channel recordings of fly song (270 pulses and sine songs), and for the noisy and variable marmoset data (1610 annotations, 400 per type), but even in these cases, the iterative training protocol can reduce the manual annotation work.

Overall, *DAS* requires small to moderate amounts of data for reaching high performance. High throughput (*Figure 4A*) and small training data sets (*Figure 4D*) translate to short training times (*Figure 4E*). The single-channel data sets typically achieve 90% of the performance after less than 10 min of training on a GPU. Training on the full data sets typically finishes in fewer than five hours. Thus, *DAS* can be adapted to novel species in short time and with little manual annotation work.

### *DAS* can be combined with unsupervised methods

*DAS* is a supervised annotation method: It discriminates syllable types that have been manually assigned different labels during training. By contrast, unsupervised methods can determine in unlabelled data whether syllables fall into distinct types and if so, classify the syllables (*Tabler et al., 2017*; *Coffey et al., 2019*; *Clemens et al., 2018*; *Goffinet et al., 2021*; *Sainburg et al., 2020*; *Sangiamo et al., 2020*; *Arthur et al., 2021*). While *DAS* does not require large amounts of manual annotations (*Figure 4D*), manual labeling of syllable types can be tedious when differences between syllable types are subtle (*Clemens et al., 2018*) or when repertoires are large (*Sangiamo et al., 2020*). In these cases, combining *DAS* with unsupervised methods facilitates annotation work. To demonstrate the power of this approach, we use common procedures for unsupervised classification, which consist of an initial preprocessing (e.g. into spectograms) and normalization (e.g. of amplitude) of the syllables, followed by dimensionality reduction and clustering (see Materials and methods) (*Clemens et al., 2018*; *Sainburg et al., 2020*; *Sangiamo et al., 2020*).

For fly song, *DAS* was trained to discriminate two major song modes, pulse and sine. However, *Drosophila melanogaster* males produce two distinct pulse types, termed $P_{slow}$ and $P_{fast}$ (*Clemens et al., 2018*), and unsupervised classification robustly discriminates the two pulse types as well as the sine song in the *DAS* annotations (*Figure 5A–C*). Mouse USVs do not fall into distinct types (*Tabler et al., 2017*; *Goffinet et al., 2021*; *Sainburg et al., 2020*). In this case, unsupervised clustering produces a low-dimensional representation that groups the syllables by the similarity of their spectrograms (*Tabler et al., 2017*; *Coffey et al., 2019*; *Sangiamo et al., 2020*, *Figure 5D,E*). For marmosets, unsupervised classification recovers the four manually defined call types (*Figure 5F,G*). However, most call types are split into multiple clusters, and the clusters for trills and twitters tend to separate poorly (*Figure 4—figure supplement 3G*), which reflects the large variability of the marmoset vocalizations. This contrasts with the song of Zebra finches, for which the unsupervised method produces a one-to-one mapping between manually defined and unsupervised syllable types (*Goffinet et al., 2021*; *Sainburg et al., 2020*; *Figure 5H,I*). For the song of Bengalese finches, the unsupervised classification recovers the manual labeling (*Goffinet et al., 2021*; *Sainburg et al., 2020*; *Figure 5J,K*) and reveals manual annotation errors: For instance, the song syllable that required >200 manual annotations to be annotated correctly by *DAS* is a mixture of three distinct syllable types (*Figure 4—figure supplement 3C–F*).

Overall, unsupervised methods simplify annotation work: *DAS* can be trained using annotations that do not discriminate between syllable types and the types can be determined *post hoc*. If distinct types have been established, *DAS* can be retrained to directly annotate these types using the labels produced by the unsupervised method as training data.

## Discussion

We here present *Deep Audio Segmenter* (*DAS*), a method for annotating acoustic signals. *DAS* annotates song in single- and multi-channel recordings from flies (*Figure 1*, *Figure 1—figure supplement 3*), mammals ( *Figure 2*, *Figure 2—figure supplement 1*), and birds (Figs *Figure 3*, *Figure 3—figure supplement 2*) accurately, robustly, and quickly (*Figure 4A,B*). *DAS* performs as well as or better than existing methods that were designed for specific types of vocalizations (*Koumura and Okanoya, 2016*; *Cohen et al., 2020*; *Tachibana et al., 2020*; *Coffey et al., 2019*, *Table 2*). *DAS* performs excellently for signals recorded on single and multiple channels (*Figure 1*, *Figure 1—figure supplement 3*), with different noise levels, and with diverse characteristics. This suggests that *DAS* is a general method for accurately annotating signals from a wide range of recording setups and species.

Using a user-friendly graphical interface, our method can be optimized for new species without requiring expert knowledge and with little manual annotation work (*Figure 4C–E*). Network performance is robust to changes in the structural parameters of the network, like filter number and duration, or the network depth (*Figure 4—figure supplement 5*). Thus, the structural parameters do *not* need to be finely tuned to obtain a performant network for a new species. We have trained networks using a wide range of signal types (*Table 4*) and these networks constitute good starting points for adapting *DAS* to novel species. We provide additional advice for the design of novel networks in Methods. This makes the automatic annotation and analysis of large corpora of recordings from diverse species widely accessible.

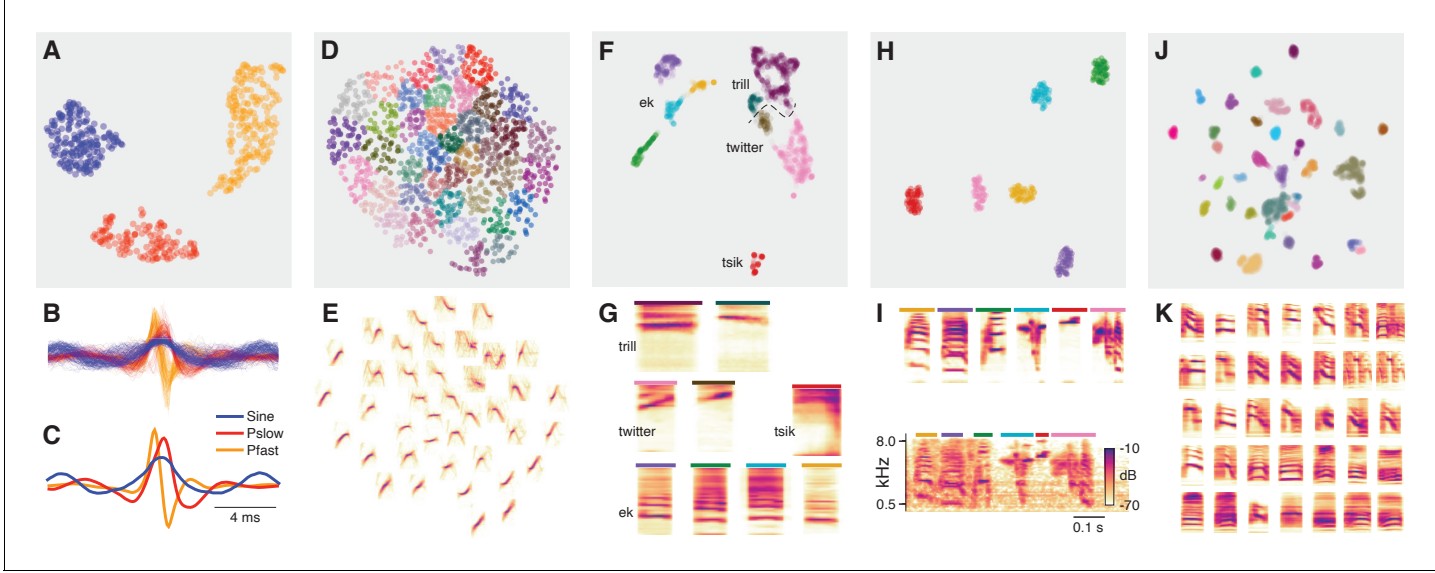

**Figure 5.** DAS can be combined with unsupervised methods for song classification. (**A**) Low-dimensional representation obtained using the UMAP (*McInnes and Healy, 2018*) method of all pulse and sine song waveforms from *Drosophila melanogaster* annotated by *DAS* in a test data set. Data points correspond to individual waveforms and were clustered into three distinct types (colors) using the density-based method HDBSCAN (*McInnes et al., 2017*). (**B, C**) All waveforms (**B**) and cluster centroids (**C**) from A colored by the cluster assignment. Waveforms cluster into one sine (blue) and two pulse types with symmetrical (red, $P_{slow}$) and asymmetrical (orange, $P_{fast}$) shapes. (**D, E**) Low-dimensional representation of the spectrograms of mouse USVs (**D**) and mean spectrogram for each cluster in D (**E**). Individual syllables (points) form a continuous space without distinct clusters. Song parameters vary continuously within this space, and syllables can be grouped by the similarity of their spectral contours using k-means clustering. (**F, G**) Low-dimensional representation of the spectrograms of the calls from marmosets (**F**) and mean spectrogram for each cluster in F (**G**). Calls separate into distinct types and density-based clustering (colors) produces a classification of syllables that recovers the manual annotations (*Figure 4—figure supplement 3G*, homogeneity score 0.88, completeness score 0.57, v-score 0.69). Most types split into multiple clusters, reflecting the variability of the call types in marmosets. Colored bars on top of each spectrogram in G correspond to the colors for the individual clusters in F. The dashed line shows the boundary separating trills and twitters. (**H, I**) Low-dimensional representation of the spectrograms of the syllables from one Zebra finch male, mean spectrogram for each cluster in H (**I**, top), and example of each clustered syllable within the motif (**I**, bottom). Density-based clustering (colors) recovers the six syllable types forming the male's motif. Colored bars on top of each spectrogram in I correspond to the colors for the individual clusters in H. (**J, K**) Low-dimensional representation of the spectrograms of the syllables from four Bengalese finch males (**J**) and mean spectrogram for each cluster in J (**K**). Syllables separate into distinct types and density-based clustering (colors) produces a classification of syllables that closely matches the manual annotations (homogeneity score 0.96, completeness score 0.89, v-score 0.92). X-axes of the average spectrograms for each cluster do not correspond to linear time, since the spectrograms of individual syllables were temporally log-rescaled and padded prior to clustering. This was done to reduce the impact of differences in duration between syllables.

We show that the annotation burden can be further reduced using unsupervised classification of syllable types, in particular for species with large or individual-specific repertoires (*Figure 5*, *Clemens et al., 2018*; *Tabler et al., 2017*; *Coffey et al., 2019*; *Goffinet et al., 2021*; *Sainburg et al., 2020*; *Arthur et al., 2021*). In the future, incorporating recent advances in the self-supervised or semi-supervised training of neural networks will likely further reduce data requirements (*Mathis et al., 2021*; *Raghu et al., 2019*; *Devlin et al., 2019*; *Chen and He, 2020*). These approaches use unlabeled data to produce networks with a general and rich representation of sound features that can then be fine-tuned for particular species or individuals using few annotated samples. *DAS* currently does not work well with recordings in which the signals produced by multiple animals overlap. In the future, *DAS* will be extended with methods for multi-speaker speech recognition to robustly annotate vocalizations from animal groups.

Lastly, the high inference speed (*Figure 4A,B*) allows integration of *DAS* in closed-loop systems in which song is detected and stimulus playback or optogenetic manipulation is triggered with low latency (*Bath et al., 2014*; *Stowers et al., 2017*). In combination with realtime pose tracking (*Mathis et al., 2018*; *Pereira et al., 2019*; *Graving et al., 2019*), *DAS* provides unique opportunities to tailor optogenetic manipulations to specific behavioral contexts, for instance to dissect the neural circuits underlying acoustic communication in interacting animals (*Coen et al., 2014*; *Fortune et al., 2011*; *Okobi et al., 2019*).

## Materials and methods

Instructions for installing and using *DAS* can be found at https://janclemenslab.org/das. Code for the `das` python module is available at https://github.com/janclemenslab/das, code for the unsupervised methods is at https://github.com/janclemenslab/das_unsupervised. All fitted models (with example data and code) can be found at https://github.com/janclemenslab/das-menagerie.

We also provide instructions for training *DAS* using google colab, which provides a GPU-accelerated python environment. Colab removes the need to install GPU libraries: Annotations can be made locally in the GUI without a GPU and training and prediction are done on GPU-accelerated nodes in the cloud. See this notebook for details: http://janclemenslab.org/das/tutorials/colab.html.

### Data sources

All data used for testing *DAS* were published previously. Sources for the original data sets, for the data and annotations used for training and testing, and for the fitted models are listed in *Table 5*. Single-channel recordings of *Drosophila melanogaster* (strain OregonR) males courting females were taken from *Stern, 2014*. The multi-channel data from *Drosophila melanogaster* (strain NM91) males courting females were recorded in a chamber tiled with nine microphones (*Coen et al., 2014*) and was previously published in *Clemens et al., 2018*. Annotations for fly song were seeded with *Fly-SongSegmenter* (*Arthur et al., 2013*; *Coen et al., 2014*) and then manually corrected. Recordings of mouse USVs were previously published in *Ivanenko et al., 2020*. The USVs were manually labeled using the *DAS* graphical interface. Marmoset recordings were taken from the data published with *Landman et al., 2020*. Since we required more precise delineation of the syllable boundaries than was provided in the published annotations, we manually fixed annotations for a subset of the data that then was used for training and testing. The network was trained and tested on a subset of four vocalization types (eks/trills/tsiks/twitters, N=603/828/115/868). The remaining vocalization types were excluded since they had 60 or fewer instances in our subset. To test *DAS* on bird songs, we used a publicly available, hand-labeled collection of song from four male Bengalese finches

**Table 5.** Sources of all data used for testing *DAS*.

'Data' refers to the data used for *DAS* and to annotations that were either created from scratch or modified from the original annotations (deposited under https://data.goettingen-research-online.de/dataverse/das). 'Original data' refers to the recordings and annotations deposited by the authors of the original publication. 'Model' points to a directory with the model files as well as a small test data set and demo code for running the model (deposited under https://github.com/janclemenslab/das-menagerie).

| Species | Reference | Data and model repositories |
|---|---|---|
| Fly single channel | *Stern, 2014* | data: https://doi.org/10.25625/TP4ODR |
| | | original data: https://www.janelia.org/lab/stern-lab/tools-reagents-data |
| | | model: https://github.com/janclemenslab/das-menagerie/dmel_single |
| Fly multi channel | *Clemens et al., 2018* | data: https://doi.org/10.25625/8KAKHJ |
| | | model: https://github.com/janclemenslab/das-menagerie/dmel_multi |
| Mouse | *Ivanenko et al., 2020* | data: https://doi.org/10.25625/VVSKCH |
| | | original data: https://data.donders.ru.nl/collections/di/dcn/DSC_620840_0003_891 |
| | | model: https://github.com/janclemenslab/das-menagerie/mouse |
| Marmoset | *Landman et al., 2020* | data: https://doi.org/10.25625/DYG3KV |
| | | original data: https://osf.io/q4bm3/ |
| | | model: https://github.com/janclemenslab/das-menagerie/marmoset |
| Bengalese finch | *Nicholson et al., 2017* | data: https://doi.org/10.25625/ENKMJS |
| | | original data: https://doi.org/10.6084/m9.figshare.4805749.v6 |
| | | model: https://github.com/janclemenslab/das-menagerie/bengalese_finch |
| Zebra finch | *Goffinet et al., 2021* | data: https://doi.org/10.25625/ZXJJJY |
| | | original data: https://research.repository.duke.edu/concern/datasets/9k41zf38g |
| | | model: https://github.com/janclemenslab/das-menagerie/zebra_finch |

(*Nicholson et al., 2017*) and recordings of female-directed song from a male Zebra finch from *Goffinet et al., 2021*. For training and testing the Zebra finch network, we manually labeled 473 syllables of six types (320 s of recordings).

## *DAS* network

*DAS* is implemented in Keras (*Chollet, 2015*) and Tensorflow (*Abadi et al., 2016*). At its core, *DAS* consists of a stack of temporal convolutional blocks, which transform an input sequence of audio data into an output sequence of labels.

### Inputs

*DAS* takes as input raw, single or multi-channel audio. Pre-processing of the audio using a wavelet or short-time Fourier transform (STFT) is optional and integrated into the network. *DAS* processes audio in overlapping chunks (*Figure 1—figure supplement 1A–D*). The chunking accelerates annotations since multiple samples are annotated in a single computational step. Edge effects are avoided by processing overlapping chunks and by discarding a number of samples at the chunk boundaries. The overlap depends on the number of layers and the duration of filters in the network.

### STFT frontend

The trainable STFT frontend is an optional step and was implemented using kapre (*Choi et al., 2017*). Each frequency channel in the output of the frontend is the result of two, one-dimensional strided convolutions which are initialized with the real and the imaginary part of discrete Fourier transform kernels:

$$x(i,f) = \sum_{\tau=0}^{T-1} x(i*s+\tau)[\cos(2\pi f\tau/T) - i\sin(2\pi f\tau/T)]$$

$$y(i,f) = \log_{10}(\sqrt{\Re x(i,f)^2 + \Im x(i,f)^2})$$

where $f$ is the frequency, $s$ is the stride, and $T$ is the filter duration. The stride results in downsampling of the input by a factor $s$.

The STFT kernels are optimized with all other parameters of the network during training. The STFT frontend was used for mammal and bird signals, but not for fly song. In the mammal and bird networks, we used 33 STFT filter pairs with a duration $T = 64$ samples and a stride $s = 16$ samples. For mouse and bird song, the STFT frontend sped up training and inference, and had a small positive impact on performance. For fly song, the STFT frontend tended to reduce performance and was omitted.

### Temporal convolutional blocks

Temporal convolutional network (TCN) blocks are central to *DAS* and produce a task-optimized hierarchical representation of sound features at high temporal resolution (*Bai et al., 2018*). Each TCN block consists of a stack of so-called residual blocks (*Figure 1—figure supplement 1E*, *He et al., 2016*):

A *dilated convolutional layer* filters the input with a number of kernels of a given duration: $y_i(t) = \sum_{\tau,\gamma} k_i(\tau,\gamma) * x(t - a\tau,\gamma)$, where $k_i(\tau,\gamma)$ is the $i$ th kernel, $x(t,\gamma)$ is the input on channel $\gamma$ at time $t$, $y_i(t)$ the output, and $a$ the gap or skip size (*Yu and Koltun, 2016*). In old-fashioned convolution $a = 1$. Increasing $a$ allows the kernel to span a larger range of inputs with the same number of parameters and without a loss of output resolution. The number of parameters is further reduced for networks processing multi-channel audio, by using *separable* dilated convolutions in the first two TCN blocks (*Mamalet and Garcia, 2012*). In separable convolutions, the full two-dimensional $k(\tau,\gamma)$ convolution over times and channels is decomposed into two one-dimensional convolutions. First, a temporal convolution, $k^t(\tau,1)$, is applied to each channel and then $N$ channel convolutions, $k^\gamma(1,\gamma)$, combine information across channels. Instead of $\tau \times \gamma$ parameters, the separable convolution only requires $\tau + N \times \gamma$ parameters. Note that each temporal convolution is applied to each channel, leading to a sharing of filter parameters across channels. This makes explicit the intuition that some

operations should be applied to all channels equally. We also tested an alternative implementation, in which individual channels were first processed separately by a single-channel TCN, the outputs of the TCN blocks for each channel were concatenated, and then fed into a stack of standard TCNs with full two-dimensional convolutions. While this architecture slightly increased performance it was also much slower and we therefore chose the architecture with separable convolutions. Architecture choice ultimately depends on speed and performance requirements of the annotation task.

A *rectifying linear unit* transmits only the positive inputs from the dilated convolutional layer by setting all negative inputs to 0: $y_i = max(0, y_i)$.

A *normalization layer* rescales the inputs to have a maximum absolute value close to 1: $y_i/(max(|y_i|) + 10^{-5})$.

The output of the residual block, $z(t)$, is then routed to two targets: First, it is added to the input: $o(t) = x(t) + z(t)$ and fed into subsequent layers. Second, via so-called skip connections, the outputs of all residual blocks are linearly combined to produce the network's final output (*van den Oord et al., 2016*).

A single TCN block is composed of a stack of five residual blocks. Within a stack, the skip size $a$ doubles - from one in the first to $2^5 = 16$ in the final residual block of a stack. This exponential increase in the span of the filter $\tau * a$ allows the TCN block to produce a hierarchical representation of its inputs, from relatively low-level features on short timescales in early stacks to more derived features on longer timescales in late stacks. Finally, a full network consists of a stack of 2 to 4 TCN blocks, which extract ever more derived features (*Figure 1—figure supplement 1A–D*).

## Output

The network returns a set of confidence scores—one for each song type (and for 'no song')—for each sample from a linear combination of the output of each residual block in the network, by using a single dense layer and a softmax activation function. Re-using information from all blocks via so-called skip connections ensures that downstream layers can discard information from upstream layers and facilitates the generation of specialized higher order presentations. If the input recording got downsampled by a STFT frontend, a final upsampling layer restores the confidence scores to the original audio rate by repeating values. The parameters of all used networks are listed in *Table 4*.

## Choice of structural network parameters

*DAS* performance is relatively robust to the choice of structural network parameters like filter duration and number, or network depth (*Figure 4—figure supplement 5*). The networks tested here are good starting points for adapting *DAS* to your own data (*Table 4*). In our experience, a network with 32 filters, filter duration 32 samples, 3 TCN blocks, and a chunk duration of 2048 samples will produce good results for most signals. A STFT downsampling layer with 32 frequency bands and 16x downsampling should be included for most signals except when the signals have a pulsatile character. These parameters have been set as defaults when creating a new *DAS* network. Given that *DAS* trains quickly (*Figure 4E*), network structure can be optimized by training *DAS* networks over a grid of structural parameters, for instance to find the simplest network (in terms of the number of filters and TCN blocks) that saturates performance but has the shortest latency. We here provide additional guidelines for choosing a network's key structural parameters:

The chunk duration corresponds to the length of audio the network processes in one step and constitutes an upper bound for the context available to the network. Choose chunks sufficiently long so that the network has access to key features of your signal. For instance, for fly song, we ensured that a single chunk encompasses several pulses in a train, so the network can learn to detect song pulses based on their regular occurrence in trains. Longer chunks relative to this timescale can reduce short false positive detections, for instance for fly sine song and for bird song. Given that increasing chunk duration does not increase the number of parameters for training, we recommend using long chunks unless low latency is of essence (see below).

Downsampling/STFT weakly affects performance but strongly accelerates convergence during training. This is because (A) the initialization with STFT filters is a good prior that reduces the number of epochs it takes to learn the optimal filters, and (B) the downsampling reduces the data bandwidth and thereby the time it takes to finish one training epoch. The overall increase in performance from adding the STFT layer is low because convolutional layers in the rest of the network can easily

replicate the computations of the STFT layer. For short pulsatile signals or signals with low sampling rates, STFT and downsampling should be avoided since they can decrease performance due to the loss of temporal resolution.

The number of TCN blocks controls the network's depth. A deeper network can extract more high-level features, though we found that even for the spectro-temporally complex song of Bengalese finches, deeper networks only weakly improved performance (*Figure 4—figure supplement 5*).

Multi-channel audio can be processed with multi-channel filters via full convolutions or with shared channel-wise filters via time-channel separable convolutions. This can be set on a per-TCN-block basis. We recommend to use separable convolutions in the first 1–2 layers, since basic feature extraction is typically the same for each channel. Later layers can then have full multi-channel filters to allow more complex combination of information across channels.

Real-time performance can be optimized by reducing networks complexity and chunk duration (*Figure 4—figure supplement 2*). We recommend starting with the default parameters suggested above and then benchmarking latency. If required, latency can then be further reduced by reducing chunk duration, the number and duration of filters, and the number of TCN blocks.

## Training

Networks were trained using the categorical cross-entropy loss and the Adam optimizer (*Kingma and Ba, 2015*) with a batch size of 32. Prediction targets were generated from fully annotated recordings and one-hot-encoded: Segments were coded as binary vectors, with $y_i(t) = 1$ if a segment of type $i$ occurred at time $t$, and $y_i(t) = 0$ otherwise. To encode uncertainty in the timing of fly song pulses, the pulses were represented as Gaussian bumps with a standard deviation of 1.6 ms. A 'no song' type was set to $y_{\mathrm{nosong}}(t) = 1 - \sum_i y_i(t)$. That way, $y$ corresponds to the probability of finding any of the annotated song types or no song. For bird song, short gaps (6.25 ms, 200 samples at 32 kHz) were introduced between adjacent syllables to aid the detection of syllable on- and offsets after inference. That way, syllable on- and offsets could be unequivocally detected as changes from 'no song' to any of the syllables. This reduced the amount of false positive on- and offsets from switches in the inferred label within a syllable.

Typically, multiple fully annotated recordings were combined in a data set. Each recording was split 80:10:10 into a training, validation, and test set. The validation and test data were randomly taken from the first, the middle or the last 10% of each recording. Given the uneven temporal distribution of call types in the marmoset recordings, we split the data 60:20:20 to ensure that each call type was well represented in each split. For all networks, training was set to stop after 400 epochs or earlier if the validation loss was not reduced for at least 20 epochs. Training typically stopped within 40–80 epochs depending on the dataset. The test set was only used after training, for evaluating the model performance.

## Generation of annotations from the network output

The confidence scores produced by the model correspond to the sample-wise probability for each song type. To produce an annotation label for each sample, the confidence scores were further processed to extract event times and syllable segments. In the resulting annotations, song types are mutually exclusive, that is, each sample is labeled as containing a single song type even if song types overlap.

Event times for event-like song types like fly pulse song were determined based on local maxima in the confidence score, by setting a threshold value between 0 and 1 and a minimal distance between subsequent peaks (using `peakutils`, *Negri and Vestri, 2017*). For the pulse song of flies, we set a minimal distance of 10 ms and a threshold of 0.7 for single channel data (*Figure 1*) and 0.5 for multi-channel data (*Figure 1—figure supplement 3*).

For segment-like song types like fly sine song or the syllables of mouse, marmoset, and bird song, we first transformed the sample-wise probability into a sequence of labels using $\mathrm{argmax}_i\, y(i, t)$. The resulting annotation of segments was then smoothed by filling short gaps (flies 20 ms, mice 10 ms, marmosets and birds 5 ms) and removing very short detections (flies 20 ms, mice 5 ms, marmosets and birds 30 ms). These values were chosen based on the statistics of song found in the training data. Syllable on- and offsets were detected as changes from no-song to song and song to no-song,

respectively. For bird and marmoset vocalizations, syllable labels were determined based on a majority vote, by calculating the mode of the sample-wise labels for each detected syllable.

## Evaluation

*DAS* was evaluated on segments of recordings that were not used during training.

### Events

For events—fly pulse song, or the on- and offsets of segments—we matched each true event with its nearest neighbor in the list of true events and counted as true positives only events within a specified distance from a true event. For the pulse song of flies as well as for the onsets and offsets of mouse, marmoset, and bird syllables, this distance was set to 10 ms. Results were robust to the specific choice of the distance threshold (*Figure 1—figure supplement 2A*). For the onsets and offsets of fly sine song and of the marmoset vocalizations, we set this distance to 40 ms, since these signals tended to fade in and out, making the delineation of exact boundaries difficult. False positive events were counted if the distance from a detected event to the nearest true event exceeded the distance threshold or if another detected event was closer to each true event within the distance threshold. If several detected pulses shared the same nearest true pulses, only the nearest of those was taken as a true positive, while the remaining detections were matched with other true pulses within the distance threshold or counted as false positives.

False negatives were counted as all true events without nearby detected events. For pulse, pseudo true negative events were estimated as the number of tolerance distances (2x tolerance distance) fitting into the recording, minus the number of pulses. These true negatives for pulse do not influence precision, recall, and F1-scores and are only used to fill the confusion matrices in *Figure 1D,I* and *Figure 1—figure supplement 3C,H*. Pulse and sine song were evaluated only up to the time of copulation.

### Matching segment labels

For songs with only one syllable type, we compared the predicted and true labels for each sample to compute the confusion matrix (*Figure 1—figure supplement 1F,G*). In the case of multiple syllable types, the mode of the true and predicted labels for the samples of each detected syllable were compared. A true positive was counted if the mode of the true labels was the same for the samples covered by the detected syllable. Using the true syllables as reference produces similar results (*Figure 2—figure supplement 1E,F* and *Figure 3—figure supplement 1D,E*).

### Performance scores

From the false negative (FN), false positive (FP), and true positive (TP) counts we extracted several scores: Precision (P)—the fraction of true positive out of all detections TP/(FP+TP)—and recall (R)—the fraction of true positives out of all positives TP/(TP+FN). The F1 score combines precision and recall via their geometric mean: $2 \times P \times R/(P + R)$. For datasets with many different syllable types, we also used as a summary measure of performance the accuracy—the fraction of correctly labelled syllables: (TP+TN)/(TP+TN+FP+FN). For comparison with other studies, we additionally provide the error rate for the song of Bengalese finches, which is based on the Levenshtein edit distance and corresponds to the minimal number of inserts, deletions, and substitutions required to transform the sequence of true syllable labels into the sequence of predicted syllable labels normalized by the length of the true sequence (*Koumura and Okanoya, 2016*; *Cohen et al., 2020*).

### Temporal precision

The temporal precision for events (pulses, syllable onsets and offsets) was calculated as the median absolute distance between all matched events.

### Annotation errors for Bengalese finches

The network for annotating bird song was trained on all syllable types. We removed from the test data one syllable type with only a single instance in the test set (which was correctly classified), because the performance could not be assessed reliably based on a single instance. We also excluded as annotation error a syllable type that contained syllables of more than six distinct types.

## Estimation of signal-to-noise ratio from audio recordings

To assess the robustness of annotation performance to noise, we assessed the recall of *DAS* for epochs with different signal-to-noise ratios (SNRs) for the fly and the mouse networks. Because of fundamental differences in the nature of the signals, SNR values were computed with different methods and are therefore not directly comparable across species.

### Pulse

Pulse waveforms were 20 ms long and centered on the peak of the pulse energy. The root-mean square (RMS) amplitudes of the waveform margins (first and last 5 ms) and center (7.5–12.5 ms) were taken as noise and signal, respectively. RMS is defined as $\sqrt{\sum_i x(i)^2}$. For multi-channel recordings, the pulse waveform from the channel with the highest center RMS was chosen to calculate the SNR.

### Sine

Signal was given by the RMS amplitudes of the recording during sine song. Noise is the RMS amplitude in the 200 ms before and after each sine song, with a 10 ms buffer. For instance, if a sine song ended at 1000 ms, the recording between 1010 and 1210 ms was taken as noise. From the 200 ms of noise, we excluded samples that were labeled as sine or pulse and included intervals between pulses. For multi-channel recordings, the SNR was calculated for the channel with the largest signal amplitude.

### Mouse

We assumed an additive noise model: $\sigma_{total}^2 = \sigma_{signal}^2 + \sigma_{noise}^2$ is the squared signal averaged over a window of 1 ms. Since noise variance changed little relative to the signal variance in our recordings, we can assume constant noise over time to calculate the signal strength: $\sigma_{signal}^2 = \sigma_{total}^2 - \sum_t \sigma_{noise}^2$. The sample-wise SNR is then given by $SNR(t) = \sigma_{signal}(t)^2 / \sum_t \sigma_{noise}^2$.

## Speed benchmarks

Inference speed was assessed using throughput and latency. Throughput is the number of samples annotated per second and latency is the time it takes to annotate a single chunk. Throughput and latency depend on the chunk duration—the duration of a recording snippet processed by the network at once—and on the batch size—the number of chunks processed during one call. Larger batches maximize throughput by more effectively exploiting parallel computation in modern CPUs and GPUs and reducing overheads from data transfer to the GPU. This comes at the cost of higher latency, since results are available only after all chunks in a batch have been processed. Using small batch sizes and short chunks therefore reduces latency, since results are available earlier, but this comes at the cost of reduced throughput because of overhead from data transfer or under-utilized parallel compute resources. To assess throughput and latency, run times of `model.predict` were assessed for batch sizes ranging from 1 to 1024 (log spaced) with 10 repetitions for each batch size after an initial warmup run (*Figure 4—figure supplement 1A–L*). Results shown in the main text are from a batch size corresponding to 1 s of recording for throughput (*Figure 4A*) and a batch size of 1 for latency (*Figure 4B*, see also *Figure 4—figure supplement 1*). For fly song, latency was optimized by reducing the chunk size to 25.6 ms (*Figure 4—figure supplement 2*). Benchmarks were run on Windows 10, Tensorflow 2.1, with the network either running on a CPU (Intel i7-7770, 3.6 GHz) or on a GPU (GTX1050 Ti 4 GB RAM).

We also benchmarked the throughput of existing methods for comparison with *DAS* (*Table 2*). Since neither of the methods considered are designed to be used in ways in which latency can be fairly compared to that of *DAS*, we did not assess latency. The throughput values include all pre-processing steps (like calculation of a spectrogram) and comparisons to *DAS* were done using the same hardware (CPU for *FSS* and *USVSEG*, GPU for *TweetyNet* and *Oikarinen et al., 2019*). The throughput of *FSS* (*Arthur et al., 2013*; *Coen et al., 2014*) was tested using 400 s of single-channel and 9-channel recordings in Matlab2019a. *USVSEG Tachibana et al., 2020* was tested on a 72 s recording in Matlab2019a. *TweetyNet* (*Cohen et al., 2020*) was tested using a set of 4 recordings (total duration 35 s). Throughput for *TweetyNet* was given by the combined runtimes of the pre-processing steps (calculating of spectrograms from raw audio and saving them as temporary files)

and the inference steps (running the network on a GPU). For the previously published network for annotating marmoset calls (*Oikarinen et al., 2019*), we relied on published values for estimating throughput: A processing time of 8 min for a 60 min recording corresponds to a throughput of 7.5 s/s.

## Data economy

For estimating the number of manual annotations required to obtain accurate annotations, we trained the networks using different fractions of the full training and validation sets (for instance, 0.001, 0.005, 0.01, 0.05, 0.1, 0.5, 1.0). Performance of all networks trained on the different subsets was evaluated on the full test sets. The number of manual annotations in each subset was determined after training from the training and validation sets. The number of annotations required to exceed 90% of the F1 score of a model trained on the full data sets was calculated based on a lowess fit (*Cleveland, 1979*) to the data points (*Figure 4A,B*).

## Unsupervised classification

Segmented signals were clustered using unsupervised methods described previously in *Clemens et al., 2018*, *Sainburg et al., 2020*, and *Sangiamo et al., 2020*. First, signals were pre-processed: For fly song, pulse and sine waveforms of duration 15 ms were extracted from the recording, aligned to their peak energy, normalized to unit norm, and adjusted for sign (see *Clemens et al., 2018* for details). For mouse, marmoset, and bird vocalizations, we adapted the procedures described in *Sainburg et al., 2020*: Noise was reduced in the bird song recordings using the `noisereduce` package (https://github.com/timsainb/noisereduce). For mouse and marmoset vocalizations, noise reduction tended to blur the spectral contours and was omitted. Then, syllable spectrograms were extracted from mel spectrograms of the recordings. The noise floor of the spectrogram at each frequency was estimated as the median spectrogram over time and each spectral band was then divided by the frequency-specific noise floor value. Finally, the spectrogram values were log-transformed and thresholded at zero for mice and two for marmosets and birds after visual inspection of the spectrograms to further remove background noise. To reduce differences in the duration of different syllables, all syllables were first log resized in time (scaling factor 8) and then padded with zeros to the duration of the longest syllable in the data set. Lastly, the frequency axis of the spectrograms for mouse syllables were aligned to the peak frequency, to make clustering robust to jitter in the frequency of the thin spectral contours (*Sangiamo et al., 2020*). The peak frequency of each mouse syllable was calculated from its time-averaged spectrogram, and only the 40 spectrogram frequencies around the peak frequency were retained.

The dimensionality of the pre-processed waveforms (fly) or spectrograms (mouse, marmoset, birds) was then reduced to two using the UMAP method (*McInnes and Healy, 2018*) (`mindist` = 0.5, 0.1 for marmosets to improve separation of clusters). Finally, signals were grouped using unsupervised clustering. For the fly, marmoset, and bird signals, the UMAP distribution revealed distinct groups of syllables and we used a density-based method to cluster the syllables (*Campello et al., 2013*, `min_samples` = 10, `min_cluster_size` = 20). For mouse USVs, no clusters were visible in the UMAP distribution and density-based clustering failed to identify distinct groups of syllables. Syllables were therefore split into 40 groups using k-means clustering.

## Open source software used

- avgn https://github.com/timsainb/avgn_paper *Sainburg et al., 2020*
- hdbscan *McInnes et al., 2017*
- ipython *Perez and Granger, 2007*
- jupyter *Kluyver et al., 2016*
- kapre *Choi et al., 2017*
- keras *Chollet, 2015*
- keras-tcn https://github.com/philipperemy/keras-tcn
- librosa *McFee et al., 2015*
- matplotlib *Hunter, 2007*
- noisereduce https://github.com/timsainb/noisereduce
- numpy *Harris et al., 2020*
- pandas *McKinney, 2010*

- peakutils *Negri and Vestri, 2017*
- scikit-learn *Pedregosa et al., 2011*
- scipy *Virtanen et al., 2020*
- seaborn *Waskom et al., 2017*
- snakemake *Köster and Rahmann, 2018*
- tensorflow *Abadi et al., 2016*
- UMAP *McInnes and Healy, 2018*
- zarr *Miles et al., 2020*
- xarray *Hoyer and Hamman, 2017*

## Acknowledgements

We thank Kurt Hammerschmidt for providing mouse data prior to publication. We thank Mala Murthy, David Stern, and all members of the Clemens lab for feedback on the manuscript.

This work was supported by the DFG through grants 329518246 (Emmy Noether) and 430158535 (SPP2205) and by the European Research Council (ERC) under the European Union's Horizon 2020 research and innovation programme (Starting Grant agreement No. 851210 NeuSoSen).

## Additional information

### Funding

| Funder | Grant reference number | Author |
| --- | --- | --- |
| Deutsche Forschungsge-meinschaft | 329518246 | Jan Clemens |
| Deutsche Forschungsge-meinschaft | 430158535 | Jan Clemens |
| European Research Council | 851210 | Jan Clemens |

The funders had no role in study design, data collection and interpretation, or the decision to submit the work for publication.

### Author contributions

Elsa Steinfath, Data curation, Validation, Writing - review and editing; Adrian Palacios-Muñoz, Julian R Rottschäfer, Deniz Yuezak, Data curation, Writing - review and editing; Jan Clemens, Conceptualization, Software, Funding acquisition, Visualization, Methodology, Writing - original draft, Project administration

### Author ORCIDs

Elsa Steinfath https://orcid.org/0000-0002-8455-9092
Adrian Palacios-Muñoz https://orcid.org/0000-0002-9335-7767
Julian R Rottschäfer https://orcid.org/0000-0003-3741-8358
Jan Clemens https://orcid.org/0000-0003-4200-8097

### Decision letter and Author response

Decision letter https://doi.org/10.7554/eLife.68837.sa1
Author response https://doi.org/10.7554/eLife.68837.sa2

## Additional files

### Supplementary files

- Transparent reporting form

## Data availability

Any code and data used during this study is deposited at https://data.goettingen-research-online.de/dataverse/das and https://github.com/janclemenslab/das (copy archived at https://archive.software-heritage.org/swh:1:rev:0cab8a136523bcfd18e419a2e5f516fce9aa4abf). All fitted models are deposited at https://github.com/janclemenslab/das-menagerie (copy archived at https://archive.software-heritage.org/swh:1:rev:c41f87f8fd77ca122ca6f2dcc4676717526aaf24).

The following dataset was generated:

| Author(s) | Year | Dataset title | Dataset URL | Database and Identifier |
|---|---|---|---|---|
| Steinfath E, Palacios-Muñoz A, Rottschäfer JR, Yuezak D, Clemens J | 2021 | Data and models for Steinfath et al. 2021 | https://data.goettingen-research-online.de/dataverse/das | Goettingen, das |

The following previously published datasets were used:

| Author(s) | Year | Dataset title | Dataset URL | Database and Identifier |
|---|---|---|---|---|
| Nicholson D, Queen JE, Sober JS | 2017 | Bengalese finch song repository | http://dx.doi.org/10.6084/m9.figshare.4805749.v5 | figshare, 10.6084/m9.figshare.4805749.v5 |
| Ivanenko A, Watkins P, Gerven MAJ, Hammerschmidt K, Englitz B | 2020 | Data from: Classifying sex and strain from mouse ultrasonic vocalizations using deep learning | https://data.donders.ru.nl/collections/di/dcn/DSC_620840_0003_891?0 | Donders Repository, di.dcn.DSC_620840_0003_891 |
| Landman R | 2017 | Data from: Close range vocal interaction through trill calls in the common marmoset (Callithrix jacchus) | https://osf.io/q4bm3/ | Open Science Framework, 10.17605/OSF.IO/PSWQD |
| Stern D | 2014 | Data from: Reported Drosophila courtship song rhythms are artifacts of data analysis. | http://research.janelia.org/sternlab/rawData.tar | Janelia, sternlab/rawData.tar |
| Goffinet J, Brudner S, Mooney R, Pearson J | 2021 | Data from: Low-dimensional learned feature spaces quantify individual and group differences in vocal repertoires | https://doi.org/10.7924/r4gq6zn8w | Duke Digital Repository, 10.7924/r4gq6zn8w |
| Clemens J, Coen P, Roemschied FA, Pereira TD, Mazumder D, Aldarondo DE, Pacheco DA, Murthy M | 2018 | Data from: Discovery of a New Song Mode in Drosophila Reveals Hidden Structure in the Sensory and Neural Drivers of Behavior. | https://doi.org/10.25625/8KAKHJ | Goettingen Research Online, 10.25625/8KAKHJ |

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
