## [Decision Letter]

**Acceptance summary:**

This paper presents and evaluates a machine learning method for segmenting and annotating animal acoustic communication signals. The paper presents results from applying the method to signals from *Drosophila*, mice, marmosets, and songbirds, but the method should be useful for a broad range of researchers who record animal vocalizations. The method appears to be easily generalizable and has high throughput and modest training times.

**Decision letter after peer review:**

Thank you for submitting your article "Fast and accurate annotation of acoustic signals with deep neural networks" for consideration by *eLife*. Your article has been reviewed by 3 peer reviewers, and the evaluation has been overseen by Ronald Calabrese as the Senior and Reviewing Editor. The following individuals involved in review of your submission have agreed to reveal their identity: Roian Egnor (Reviewer #1); Todd Troyer (Reviewer #3).

Essential revisions:

There are two main concerns with this paper that all three reviewers shared in their own way. The reviews offer details of how to address these concerns.

1. The authors claim that the method compares favorably to other machine learning methods, but they only provide head-to-head performance comparisons for *Drosophila* songs. There are no direct comparisons against other methods for mouse ultrasonic or songbird vocalizations, nor is there a readable summary of performance numbers from the literature. This makes it difficult for the reader to assess the authors' claim that DeepSS is a significant advance over the state of the art.

2. The authors provide little discussion about optimizing network parameters for a given data set. If the software is to be useful for the non-expert, a broader discussion of considerations for setting parameters would be useful. How should one choose the stack number, or the size or number of kernels? Moreover, early in the paper DeepSS is claimed as a method that learns directly from the raw audio data, in contrast to methods that rely on Fourier or Wavelet transforms. Yet in the Methods section it is revealed that a short-term Fourier front-end was used for both the mouse and songbird data.

*Reviewer #1 (Recommendations for the authors):*

General comments:

This is a reasonable piece of software. Not super polished, but pretty straightforward. Once I got it installed (which was a little involved) it worked out of the box on raw mouse vocalizations. There could be a little more background on other methods/approaches and their strengths and weaknesses. In addition, I think more explanation about tuning parameters is necessary if the goal is for users without machine learning expertise to be able to use this.

I. Results

A. How good is it?

1. How fast is it?

a. How long to train?

Train time depends on the amount of data, but the ranges quotes (10 minutes to 5 hours) are quite reasonable. It works on reasonable hardware (I tested on a laptop with a GPU).

b. How long to classify?

Latency to classification is between 7-15ms, which is a little long for triggered optogenetics, but not bad, and certainly reasonable for acoustic feedback.

2. How accurate is it?

a. Accuracy is improved relative to Fly Song Segmenter, particularly in recall (Arthur et al., 2013; Coen et al., 2014).

Pulse song:

DeepSS precision: 97%, recall: 96%

FlySongSegmenter: precision: 99%, recall 87%.

Sine song:

DeepSS precision: 92%, recall: 98%

FlySongSegmenter: precision: 91%, recall: 91%.

b. One main concern I have is that all the signals described, with the exception of pulse song, are relatively simple tonally. Bengalese finch song is much less noisy than zebra finch song. Mouse vocalizations are quite tonal. How would this method work on acoustic signals with noise components, like zebra finches or some non-human primate signals? Some signals can have variable spectrotemporal structure based on the distortion due to increased intensity of the signal (see, for example, Fitch, Neubauer, and Hertzel, 2002).

W.Tecumseh Fitch, Jürgen Neubauer and Hanspeter Herzel (2002) "Calls out of chaos: the adaptive significance of nonlinear phenomena in mammalian vocal production" Animal Behaviour, 63: 407-418. doi:10.1006/anbe.2001.1912

B. How easy to use?

0. "our method can be optimized for new species without requiring expert knowledge and with little manual annotation work." There isn't a lot of explanation, either in the paper or in the associated documentation, of how to select network parameters for a new vocalization type. However, it does appear that small amounts of annotation are sufficient to train a reasonable classifier.

1. How much pre-processing of signals is necessary?

All the claims of the paper are based on pre-processed audio data, although they state, in the Methods section that preprocessing is not necessary. It's not clear how important this pre-processing is for achieving the kinds of accuracy observed. Certainly I would expect the speed to drop if high frequency signals like mouse vocalizations aren't downsampled. However, I tried it on raw, un-preprocessed mouse vocalizations, without downsampling and using very few training examples, and it worked quite well, only missing low signal-to-noise vocalizations.

C. How different from other things out there?

It would strengthen the paper to include some numbers on other mouse and birdsong methods, rather than simple and vague assertions "These performance values compare favorably to that of methods specialized to annotate USVs (Coffey et al., 2019; Tachibana et al., 2020; Van Segbroeck et al., 2017)." "Thus, DeepSS performs as well as or better than specialized deep learning-based methods for annotating bird song (Cohen et al., 2020; Koumura and Okanoya, 2016).

D. Miscellaneous comments

1. Interestingly, the song types don't appear to be mutually exclusive. One can have pulse song in the middle of sine song. That might be useful to be able to toggle…I can imagine cases where it would be nice to be able to label things that overlap, but in general if something is sine song, it can't be pulse song. And my assumption certainly was that song types would be mutually exclusive. Adding some explanation of that to the text/user's manual would be useful.

2. How information is combined across channels is aluded to several times but not described well in the body of the manuscript, though it is mentioned in the methods in vague terms:

"several channel convolutions, 𝑘𝛾(1, 𝛾), combine information across channels."

II. Usability

A. Getting it installed

Installing on Windows 10, was a bit involved if you were not already using python: Anaconda, python, tensorflow, CUDA libraries, create an account to download cuDNN, and update NVIDIA drivers.

However, it went OK until I hit this:

error:

File "…\miniconda3\envs\dss\lib\ctypes\__init__.py", line 373, in __init__

self._handle = _dlopen(self._name, mode)

FileNotFoundError: Could not find module '…\miniconda3\envs\dss\lib\site-packages\scipy\.libs\libbanded5x.3OIBJ6VWWPY6GDLEMSTXSIPCHHWASXGT.gfortran-win_amd64.dll' (or one of its dependencies). Try using the full path with constructor syntax.

rummaging around in Stackoverflow I found this (answer #4):

https://stackoverflow.com/questions/59330863/cant-import-dll-module-in-python

which was to edit ~\Miniconda3\envs\dss\lib\ctypes\ _init_.py to change winmode=None to winmode=0.

This worked, but did slow me down a fair bit. I'm not sure who the target audience is for this software, but it may be a bit much for the average biologist.

B. Using it

It wasn't clear to me what " complete annotation" meant: "Once you have completely annotated the song in the first 18 seconds of the tutorial recording…". Does this mean that all instances of pulse and sine song must be labeled? What are the consequences if this is not true?

I also wasn't clear how to know when it finished training and I could try predicting.

I was a bit confused by there being 2 menu headings called "DeepSS", one of which (the rightmost) has a Predict option and one of which doesn't. The first time through I used the leftmost DeepSS menu heading for Make dataset and Train, and it didn't work, the second time through I used the rightmost one and it did work (but this might have been user error).

III. General usability:

1. It would be nice if the oscillogram vertical axis didn't autoscale, it made it hard to recognize sine song when there were no pulses.

2. It would be nice to be able to change the playback rate, or the size of the jumps in the navigation bar at the bottom.

3. It was not very clear in the instructions how to fix errors and use those annotations to train again, although in the body of the text this is explicitly stated as important.

"We find that DeepSS can be efficiently trained using an iterative protocol (Pereira et al., 2018) (Figure 4C, S8): Annotate a small set of recordings and train the network for a few epochs; then generate annotations on a larger set of recordings and correct these annotations."

"Correct any prediction errors-add missing annotations, remove false positive annotations, adjust the timing of annotations."

4. The instructions for correcting could be more clear. It took a little fiddling to figure out I needed to switch to "add pulse_proposals" in order to remove a pulse proposal.

*Reviewer #2 (Recommendations for the authors):*

In general, I recommend publication after major revision.

Apart from my comments above, my excitement for this toolbox is a little bit hampered by a few aspects:

Regarding the comparisons to other tools, it would be great to compare (or at least discuss the performance of other recent tools on the bird and mouse dataset). Existing models are described as slow (Line 50), but no quantitative comparisons are given.

What are the human recall/precision for Figure 1 (compare to MARS, where a nice classifier vs. human comparison is given: https://www.biorxiv.org/content/10.1101/2020.07.26.222299v1). Is FSS a tunable algorithm and how was it tuned?

While the authors emphasize that the architecture does not require any pre-processing, they end up using the "optional" Short-time Fourier transform frontend for all, but the sine wave detection for flies (i.e. mice and bird). So I think this emphasis is ill advised.

In the end, the authors use different network parameters for all datasets (see Table S1A; with/without STFT, # stacks, kernel size…). This begs the question is this necessary. What happens if one trains the same architectures on the different datasets, does the performance suffer? -- I would suppose that is not the case, and with respect to the usability of this toolbox this seems to be an important question. I.e. how should users pick the architecture and hyperparameters?

*Reviewer #3 (Recommendations for the authors):*

The most important issue I had with the paper is that it provided the reader with little sense of DeepSS's performance relative to other software. The paper would be much better if other approaches were implemented on publicly available data sets and compared head-to-head.

The methods should state very clearly where the data sets used are available for download. While the authors may believe in DeepSS, other challengers will inevitably arise. These authors should be given the opportunity to report head-to-head performance against DeepSS on the same data.

The text references in the mouse section are to figure 3, but the relevant figure is now figure 2.

There are several places where the color scale used is useless, namely in the confusion matrices (Figure 1D, 1I, 2E, 2H, 3C and some supplemental figs). At these performance levels, color just indicates the correct vs. incorrect parts of the graph – relative shades of the heat map are not perceptible.

The birdsong confusion matrices are similarly uninformative, since all that is seen is a bunch of boxes along the diagonal. Worse yet, the discretization of the figure makes it appear that boxes along the diagonal have different sizes/offsets, suggesting structure where there is none. Showing marginals of the matrix like Figure S4E can be useful.

In Figure S4C, precision and recall are plotted whereas in S4E false positive and false negatives are plotted. Are both plots needed?

---

## [Author Response]

Essential revisions:There are two main concerns with this paper that all three reviewers shared in their own way. The reviews offer details of how to address these concerns.1. The authors claim that the method compares favorably to other machine learning methods, but they only provide head-to-head performance comparisons for *Drosophila* songs. There are no direct comparisons against other methods for mouse ultrasonic or songbird vocalizations, nor is there a readable summary of performance numbers from the literature. This makes it difficult for the reader to assess the authors' claim that DeepSS is a significant advance over the state of the art.

We addressed this issue in two ways: By testing DAS on two more species and by directly comparing accuracy and speed of DAS to that of other methods.

First, we included two more species – Zebra finches and marmosets – to further demonstrate the versatility and robustness of DAS. Both species were suggested by Reviewer #1 because they produced more variable signals. Annotations for the marmoset calls are particularly challenging because of a lot of background noise in the recordings that we used. We find that DAS annotates signals from both species with high accuracy. The results of these tests are now discussed in two new sections Results and shown in two new Figure 2 —figure supplement 1 (marmoset) and Figure 3 —figure supplement 2 (zebra finch).

For the marmosets (Results, p6, l200):

“Marmoset vocalizations

We next examined the robustness of annotations produced by DAS to noisy recordings and variable vocalization types, by training a network to annotate vocalization from pairs of marmosets (Landman et al., 2020). The recordings contain lots of background noises like faint calls from nearby animals, overdrive from very loud calls of the recorded animals, and large variability within syllable types (Figure 2 —figure supplement 1A-D). Recently, a deep-learning based method was shown to produce good performance (recall 77%, precision 85%, 12.5 ms temporal error) when trained on 16000 syllables to recognize seven vocalization types (Oikarinen et al., 2019). We trained DAS on 1/9th of the data (1800 syllables) containing 4 of the 7 vocalization types. Despite the noisy and variable vocal- izations, DAS achieves high syllable-wise precision and recall (96%, 92%, (Figure 2 —figure supplement 1E, F)). Note that DAS obtains this higher performance at millisecond resolution (temporal error 4.4 ms, Figure 2 —figure supplement 1G), while the method by Oikarinen et al., (2019) only produces annotations with a resolution of 50 ms (Table 2).”

For the Zebra finches (Results, p7, l232):

“To further demonstrate the robustness of DAS, we trained a network to annotate song from Zebra finches. In Zebra finch males, Individual renditions of a given syllable type tend to be more variable (Fitch et al., 2002). Moreover, the particular recordings used here (Goffinet et al., 2021) contain background noise from the bird’s movement. Despite the variability and noise, DAS annotates the six syllables from a male’s main motif with excellent precision and recall, and low temporal error, demonstrating that DAS is robust to song variability and recording noise (Figure 3 —figure supplement 2).”

These two species are now also included in all analyses of the throughput and latency, of number of annotations required for training, and of the unsupervised classification of syllable types. With the inclusion of these two new species, we now demonstrate state-of-the-art performance for DAS using recordings from five different species with highly diverse signal structures and signal-to-noise ratios – including the pulse song of *Drosophila*, the spectrotemporally complex syllables of birds, and noisy recordings of the variable calls of marmosets. We are not aware of a single method with demonstrably good performance on such a diverse set of annotation tasks.

Second, we compared the annotation performance and speed of our method to those of existing state-of-the-art methods for *all* species tested. This confirms our statement in the original paper: that DAS performs as well as or better than existing, specialized methods. DAS performance for mouse (against USVSEG) and bird song (against TweetyNet) is on par with that of the state of the art. DAS outperforms the state of the art for marmoset (against the Deep Learning method from Oikarinen et al., (2019)) and fly song (against FlySongSegmenter). DAS has also 3x-20x higher throughput than these methods. Overall, this established DAS as a fast, accurate, and universal method for annotating acoustic signals.

We revised the wording throughout the manuscript to more accurately reflect the outcome of these comparisons. For instance, in the comparison with bird song we now state that our method “performs as well as” the state of the art.

We added a new paragraph, describing the results of the speed comparisons (Results, p8, l264):

“We also compared the speed of DAS to that of other methods that were specifically developed to annotate the types of signals tested here. Since most existing methods are not suitable for estimating latency due to constraints in their design and interface, we only compared throughput. We find that DAS achieves 3x to 10x higher throughput than existing methods (Table 2). This has three main reasons: First, the relatively simple, purely convolutional architecture exploits the parallel processing capabilities of modern CPUs and GPUs. Second, Fourier or wavelet-like preprocessing steps are integrated into the network and profit therefore from a fast implementation and hardware acceleration. Third, for multi-channel data, DAS combines information from all audio channels early, which increases throughput by reducing the data bandwidth.

Overall, DAS annotates audio with high throughput (>8x realtime) and low latency (<15 ms), and is faster than the alternative methods tested here. The high speed renders suitable for annotating large corpora and for realtime applications without requiring specialized hardware.”

2. The authors provide little discussion about optimizing network parameters for a given data set. If the software is to be useful for the non-expert, a broader discussion of considerations for setting parameters would be useful. How should one choose the stack number, or the size or number of kernels? Moreover, early in the paper DeepSS is claimed as a method that learns directly from the raw audio data, in contrast to methods that rely on Fourier or Wavelet transforms. Yet in the Methods section it is revealed that a short-term Fourier front-end was used for both the mouse and songbird data.

We addressed this issue in two ways: First, by demonstrating that performance is relatively robust to changes in network parameters. Second, by providing advice on how to choose network parameters in Methods.

First, we performed parameter sweeps for the networks trained on the song of flies and of Bengalese finches demonstrating that DAS performance is relatively robust to changes in individual network parameters, like the number of stacks, the number/duration of filters (new Figure 4 —figure supplement 5). This means that the exact choice of parameters is not crucial to obtain performant networks. The parameters of the networks tested in the manuscript serve as good starting points for fitting networks to new species.

Second, we added a new section to Methods in which we provide advice for choosing the structural parameters of the network for a new species. The results from the parameter sweeps and the new section in Methods are now referred to in Discussion (p11, l355):

“Network performance is robust to changes in the structural parameters of the network, like filter number and duration, or the network depth (Figure 4 —figure supplement 5). Thus, the structural parameters do not need to be finely tuned to obtain a performant network for a new species. We have trained networks using a wide range of signal types (Table 4) and these networks constitute good starting points for adapting DAS to novel species. We provide additional advice for the design of novel network in Methods.”

From Methods (new section on p15, l475):

“Choice of structural network parameters

DAS performance is relatively robust to the choice of structural network parameters like filter duration and number, or network depth (Figure 4 —figure supplement 5). […] If required, latency can then be further reduced by reducing chunk duration, the number and duration of filters, and the number of TCN blocks.”

Overall, this shows that DAS is easy to adapt to novel species: annotation performance does not crucially depend on the networks’ structural parameters, and that defaults given in the new section in Methods perform well across species in our experience.

Moreover, early in the paper DeepSS is claimed as a method that learns directly from the raw audio data, in contrast to methods that rely on Fourier or Wavelet transforms. Yet in the Methods section it is revealed that a short-term Fourier front-end was used for both the mouse and songbird data.

We agree that our wording was a misleading. In DAS, raw audio is provided as an input to the network, and a trainable downsampling layer which is initialized with STFT filters can be optionally added. This has several advantages

1. The downsampling can be omitted. This makes DAS more flexible, since it allows DAS to annotate signals for which an STFT-like transform is not appropriate (fly pulse song).

2. The downsampling is initialized with STFT filters but is then trained, removing the need to hand-tune this step and allowing the network to learn filters that greatly deviate from the initialization.

3. By integrating downsampling into the network, it profits from an efficient implementation and hardware acceleration, increasing throughout and reducing latency.

We have de-emphasized and clarified this point in the beginning of Results (p3, l95):

“… First, the pre-processing step is optional. This makes DAS more flexible, since signals for which a time-resolved Fourier transform is not appropriate—for instance, short pulsatile signals—can now also be processed. Second, the optional preprocessing step is integrated and optimized with the rest of the network. This removes the need to hand-tune this step and allows the network to learn a preprocessing that deviates from a time-resolved Fourier or wavelet transform if beneficial (Choi et al., 2017). Integrating the preprocessing into the network also increases inference speed due to the efficient implementation and hardware acceleration of deep-learning frameworks.”

And in Methods (p13, l414)

“DAS takes as input raw, single or multi-channel audio. Pre-processing of the audio using a Wavelet or short-time Fourier transform is optional and integrated into the network.”

Reviewer #1 (Recommendations for the authors):This is a reasonable piece of software. Not super polished, but pretty straightforward. Once I got it installed (which was a little involved) it worked out of the box on raw mouse vocalizations. There could be a little more background on other methods/approaches and their strengths and weaknesses. In addition, I think more explanation about tuning parameters is necessary if the goal is for users without machine learning expertise to be able to use this.

We now discuss key architectural differences to existing approaches in Results (p2, l82):

“These [AM and FM] patterns are typically made explicit using a hand-tuned pre-processing step based on time-resolved Fourier or wavelet transforms (Arthur et al., 2013; Coffey et al., 2019; Cohen et al., 2020a; Oikarinen et al., 2019; Van Segbroeck et al., 2017). Most deep-learning based methods then treat this pre-defined spectrogram as an image and use methods derived from computer vision to extract the AM and FM features relevant for annotation (Coffey et al., 2019; Cohen et al., 2020a; Oikarinen et al., 2019). Recurrent units are sometimes used to track the sound features over time (Cohen et al., 2020a). This approach can produce accurate annotations but has drawbacks: First, the spectrogram constitutes a strong and proven pre-processing step, but it is unsuitable for some signal types, like short pulsatile signals. Second, the pre-processing transform is typically tuned by hand and may therefore require expert knowledge for it to produce optimal results. Lastly, the recurrent layers used in some methods (Cohen et al., 2020a) excel at combining information over time to provide the context information necessary to annotate spectrally complex signals, but they can be hard to train and slow to run (Bai et al., 2018).”

I. ResultsA. How good is it?1. How fast is it?a. How long to train?Train time depends on the amount of data, but the ranges quotes (10 minutes to 5 hours) are quite reasonable. It works on reasonable hardware (I tested on a laptop with a GPU).b. How long to classify?Latency to classification is between 7-15ms, which is a little long for triggered optogenetics, but not bad, and certainly reasonable for acoustic feedback.2. How accurate is it?a. Accuracy is improved relative to Fly Song Segmenter, particularly in recall (Arthur et al., 2013; Coen et al., 2014).Pulse song:DeepSS precision: 97%, recall: 96%FlySongSegmenter: precision: 99%, recall 87%.Sine song:DeepSS precision: 92%, recall: 98%FlySongSegmenter: precision: 91%, recall: 91%.b. One main concern I have is that all the signals described, with the exception of pulse song, are relatively simple tonally. Bengalese finch song is much less noisy than zebra finch song. Mouse vocalizations are quite tonal. How would this method work on acoustic signals with noise components, like zebra finches or some non-human primate signals? Some signals can have variable spectrotemporal structure based on the distortion due to increased intensity of the signal (see, for example, Fitch, Neubauer, and Hertzel, 2002).W.Tecumseh Fitch, Jürgen Neubauer and Hanspeter Herzel (2002) "Calls out of chaos: the adaptive significance of nonlinear phenomena in mammalian vocal production" Animal Behaviour, 63: 407-418. doi:10.1006/anbe.2001.1912

See Essential Revision #1 for details.

We tested DAS on signals from Zebra finches and marmosets. The marmoset recordings are particularly challenging due to high background noise. DAS annotates the signals from both species accurately: F1 score of 94% for marmosets (new Figure 2 —figure supplement 1, reproduced on page 3), and of 100% for the main motif of a Zebra finch (new Figure 3 —figure supplement 2, reproduced on page 4).

B. How easy to use?0. "our method can be optimized for new species without requiring expert knowledge and with little manual annotation work." There isn't a lot of explanation, either in the paper or in the associated documentation, of how to select network parameters for a new vocalization type. However, it does appear that small amounts of annotation are sufficient to train a reasonable classifier.

See Essential Rev #2 for details.

Using a parameter sweep for the fly and Bengalese finch networks, we show that DAS performance is relatively robust to changes in individual parameters – so exact parameter values do not matter (Figure 4 —figure supplement 5, reproduced on page 7). We provide recommendations for parameter choice in a new section in Methods (p15, l475, reproduced on page 7).

1. How much pre-processing of signals is necessary?All the claims of the paper are based on pre-processed audio data, although they state, in the Methods section that preprocessing is not necessary. It's not clear how important this pre-processing is for achieving the kinds of accuracy observed. Certainly I would expect the speed to drop if high frequency signals like mouse vocalizations aren't downsampled. However, I tried it on raw, un-preprocessed mouse vocalizations, without downsampling and using very few training examples, and it worked quite well, only missing low signal-to-noise vocalizations.

See Essential Rev #2.

We clarified the wording in the beginning of the Results section. We de-emphasized the misleading statement that no there is no pre-processing (Results, p3, l95, reproduced on page 9).

The pre-processing is not a manual step but an optional part of the network, which is recommended for most signals. It acts as a down-sampling layer, which is in essence a convolutional layer that is initialized with STFT filters. These filters are optimized during training. This abrogates the need for manually tuning the STFT parameters as in other methods. We did not use it for fly song, since (1) an STFT-like transform is not appropriate for the highly transient pulses in pulse song and (2) the audio rate was already low (10 kHz)–downsampling would have impaired temporal accuracy.

And yes, the pre-processing has only a small impact on performance (we observed this with mouse and bird song). We find that it accelerates convergence during training because the STFT filters are a good initialization and it accelerates inference because the downsampling reduces the bandwidth of the data processed in the network.

The above considerations are now included in our recommendation for choosing model parameters (Methods, p15, l495):

“Downsampling/STFT weakly affects performance but strongly accelerates convergence during training. This is because (A) the initialization with STFT filters is a good prior that reduces the number of epochs it takes to learn the optimal filters, and (B) the downsampling reduces the data bandwidth and thereby the time it takes to finish one training epoch. The overall increase in performance from adding the STFT layer is low because convolutional layers in the rest of the network can easily replicate the computations of the STFT layer. For short pulsatile signals or signals with low sampling rates, STFT and downsampling should be avoided since they can decrease performance due to the loss of temporal resolution.”

C. How different from other things out there?It would strengthen the paper to include some numbers on other mouse and birdsong methods, rather than simple and vague assertions "These performance values compare favorably to that of methods specialized to annotate USVs (Coffey et al., 2019; Tachibana et al., 2020; Van Segbroeck et al., 2017)." "Thus, DeepSS performs as well as or better than specialized deep learning-based methods for annotating bird song (Cohen et al., 2020; Koumura and Okanoya, 2016).

See Essential Rev #1 for details (pages 1-6 of the reply).

See Essential Revisions #1 for details (pages 1-6 of the reply).

We now compared the accuracy and speed of DAS to that of other tools for mice (USVSEG), Bengalese and Zebra finches (TweetyNet), and marmosets (Oikarinen et al., 2019). DAS is as good as or better then these specialized methods, and between 3x and 20x faster. The comparisons are shown in a new Table 2 (reproduced on page 5).

We have also tested DAS on two more species – marmosets and Zebra finches. For both species, DAS performs excellently (F1 scores 94% for marmosets and 100% for Zebra finches). The two new figures and the two new sections from Results are reproduced on pages 2-4.

Our wording is now more accurate throughout, to reflect the outcome of these comparisons. For instance, when discussing our results on bird song (p7, l229):

“Thus, DAS performs as well as specialized deep learning-based methods for annotating bird song (Cohen et al., 2020a; Koumura and Okanoya, 2016) (Table 2).”

D. Miscellaneous comments1. Interestingly, the song types don't appear to be mutually exclusive. One can have pulse song in the middle of sine song. That might be useful to be able to toggle…I can imagine cases where it would be nice to be able to label things that overlap, but in general if something is sine song, it can't be pulse song. And my assumption certainly was that song types would be mutually exclusive. Adding some explanation of that to the text/user's manual would be useful.

Yes, song types are mutually exclusive at the moment – we currently choose labels using an argmax operation over the confidence scores for each song type. In a future version one could detect overlapping song types by using an absolute threshold on the confidence scores or by training type-specific networks. We now make the fact that labels are mutually exclusive explicit

In Results (p3, l112):

“Annotation labels for the different song types are mutually exclusive and are produced by comparing the confidence score to a threshold or by choosing the most probable song type.”

And in Methods (p16, l538):

“In the resulting annotations, song types are mutually exclusive, that is, each sample is labelled as containing a single song type even if song types overlap.”

2. How information is combined across channels is aluded to several times but not described well in the body of the manuscript, though it is mentioned in the methods in vague terms:"several channel convolutions, ky(1,y), combine information across channels."

This is learned by the network during training: Different audio channels are treated like different color channels are treated in vision-based networks – they are filtered by multi-channel filters and the filter weights determine how information is combined across channels. There is no manual step in which information is explicitly combined across channels. In addition to the technical description in the methods, this is now also stated in Results when describing the network multi-channel recordings of fly song (p5, l170):

“DAS processes multi-channel audio by using filters that take into account information from all channels simultaneously.”

II. UsabilityA. Getting it installedInstalling on Windows 10, was a bit involved if you were not already using python: Anaconda, python, tensorflow, CUDA libraries, create an account to download cuDNN, and update NVIDIA drivers.

Indeed, getting a GPU accelerated version of tensorflow running is quite involved atm. Installing the CUDA and cuDNN libraries using conda is currently broken. Instead, we now provide instructions for training and inference on colab, which provides server-based GPU instances – see https://janclemenslab.org/das/tutorials/colab.html. This is now mentioned in methods: (p13, l386):

“We also provide instructions for training DAS using google colab, which provides a GPU-accelerated python environment for the network. Colab removes the need to install GPU libraries: Annotations can be made locally in the GUI without a GPU and training and predicting is done on fast GPU nodes in the cloud. See this notebook for more details: [http://janclemenslab.org/das/tutorials/colab.html].”

However, it went OK until I hit this:error:File "…\miniconda3\envs\dss\lib\ctypes\__init__.py", line 373, in __init__self._handle = _dlopen(self._name, mode)FileNotFoundError: Could not find module '…\miniconda3\envs\dss\lib\site-packages\scipy\.libs\libbanded5x.3OIBJ6VWWPY6GDLEMSTXSIPCHHWASXGT.gfortran-win_amd64.dll' (or one of its dependencies). Try using the full path with constructor syntax.rummaging around in Stackoverflow I found this (answer #4):https://stackoverflow.com/questions/59330863/cant-import-dll-module-in-pythonwhich was to edit ~\Miniconda3\envs\dss\lib\ctypes\ _init_.py to change winmode=None to winmode=0.This worked, but did slow me down a fair bit. I'm not sure who the target audience is for this software, but it may be a bit much for the average biologist.

Sorry about that. We test using CI on linux/OSX/windows and use the software in the lab on all three OSes without problems. We never encountered this particular issue. We have created a conda package (https://anaconda.org/ncb/das), which should make the installation of a set of inter-compatible packages more reliable.

B. Using itIt wasn't clear to me what " complete annotation" meant: "Once you have completely annotated the song in the first 18 seconds of the tutorial recording…". Does this mean that all instances of pulse and sine song must be labeled? What are the consequences if this is not true?

Yes, all instances in the segment must be annotated for training. Unlabelled instances would constitute false negatives in the training data and reduce performance.

This is now clarified as meaning “all pulses and sine song segments in this stretch of the recording” in the documentation.

I also wasn't clear how to know when it finished training and I could try predicting.

Training is now indicated using a dialog window in the GUI which will close once training is finished and prediction can start.

I was a bit confused by there being 2 menu headings called "DeepSS", one of which (the rightmost) has a Predict option and one of which doesn't. The first time through I used the leftmost DeepSS menu heading for Make dataset and Train, and it didn't work, the second time through I used the rightmost one and it did work (but this might have been user error).

This is now fixed.

III. General usability:

Thank you for testing the software so thoroughly and for the constructive feedback!

1. It would be nice if the oscillogram vertical axis didn't autoscale, it made it hard to recognize sine song when there were no pulses.

Sine can indeed be hard to detect when there are pulses or when there is low noise. In those cases, we use the spectrogram view to annotate sine. We will add manual vertical scaling to a future version of the software.

2. It would be nice to be able to change the playback rate, or the size of the jumps in the navigation bar at the bottom.

We added a scrollbar and a text field for entering the time to jump to for more easily navigating long recordings.

3. It was not very clear in the instructions how to fix errors and use those annotations to train again, although in the body of the text this is explicitly stated as important."We find that DeepSS can be efficiently trained using an iterative protocol (Pereira et al., 2018) (Figure 4C, S8): Annotate a small set of recordings and train the network for a few epochs; then generate annotations on a larger set of recordings and correct these annotations.""Correct any prediction errors-add missing annotations, remove false positive annotations, adjust the timing of annotations."

The documentation now contains more information on how to do that:

https://janclemenslab.org/das/quickstart.html#proof-reading

4. The instructions for correcting could be more clear. It took a little fiddling to figure out I needed to switch to "add pulse_proposals" in order to remove a pulse proposal.

Thanks! We have updated the documentation accordingly:

https://janclemenslab.org/das/quickstart.html#proof-reading

Reviewer #2 (Recommendations for the authors):In general, I recommend publication after major revision.Apart from my comments above, my excitement for this toolbox is a little bit hampered by a few aspects:Regarding the comparisons to other tools, it would be great to compare (or at least discuss the performance of other recent tools on the bird and mouse dataset). Existing models are described as slow (Line 50), but no quantitative comparisons are given.

See Essential Revisions #1 for details (pages 1-6 of the reply).

We now compared the accuracy and speed of DAS to that of other tools for mice (USVSEG), Bengalese and Zebra finches (TweetyNet), and marmosets (Oikarinen et al., 2019). DAS is as good as or better then these specialized methods, and between 3x and 20x faster. The comparisons are shown in a new Table 2 (reproduced on page 5).

We have also tested DAS on two more species – marmosets and Zebra finches. For both species, DAS performs excellently (F1 scores 94% for marmosets and 100% for Zebra finches). The two new figures and the two new sections from Results are reproduced on pages 2-4.

Our wording is now more accurate throughout, to reflect the outcome of these comparisons. For instance, when discussing our results on bird song (p7, l229):

“Thus, DAS performs as well as specialized deep learning-based methods for annotating bird song (Cohen et al., 2020a; Koumura and Okanoya, 2016) (Table 2). “

What are the human recall/precision for Figure 1 (compare to MARS, where a nice classifier vs. human comparison is given: https://www.biorxiv.org/content/10.1101/2020.07.26.222299v1).

To estimate human recall/precision, two humans annotated the test data for fly song. The results are presented in a new Table 3.

Discrepancies between human annotators arise largely in the boundaries of sine song (DAS also struggles with these) and whether isolated pulses are counted or not. DAS is as good as or better than FlySongSegmenter and human annotators. Human recall and precision are now marked in Figures 1E and J and addressed in Results (p5, l152):

“A comparison of DAS’ performance to that of human annotators reveals that our methods exceeds human-level performance for pulse and sine (Figure 1E, J, Table 3).”

Is FSS a tunable algorithm and how was it tuned?

We tested the version of FSS currently used by several labs (found at https://github.com/murthylab/MurthyLab_FlySongSegmenter). This version was hand-tuned for optimal performance for the single and multi-channel recordings used here and is used in several publications (Coen et al., 2014, 2016, Clemens et al., 2018).

While the authors emphasize that the architecture does not require any pre-processing, they end up using the "optional" Short-time Fourier transform frontend for all, but the sine wave detection for flies (i.e. mice and bird). So I think this emphasis is ill advised.

See Essential Revisions #2 for details (page 9 of the reply).

We now de-emphasize and clarify this point. The crucial difference to existing methods, in which the Short-time Fourier transform (STFT) is a mandatory and separate pre-processing step that is tuned manually, DAS makes this step optional, integrates it into the network, and optimizes the STFT filters with the rest network.

In the end, the authors use different network parameters for all datasets (see Table S1A; with/without STFT, # stacks, kernel size…). This begs the question is this necessary. What happens if one trains the same architectures on the different datasets, does the performance suffer? -- I would suppose that is not the case, and with respect to the usability of this toolbox this seems to be an important question. I.e. how should users pick the architecture and hyperparameters?

See Essential Revisions #2 for details (pages 6-8 of the reply).

We now demonstrate that DAS performance is fairly robust to changes in individual parameters (so exact parameter choices do not matter, Figure 4 —figure supplement 5), recommend default parameters that should produce performant networks for any type of signal and provide additional details on parameter choices (Methods p15, l475).

Reviewer #3 (Recommendations for the authors):The most important issue I had with the paper is that it provided the reader with little sense of DeepSS's performance relative to other software. The paper would be much better if other approaches were implemented on publicly available data sets and compared head-to-head

See Essential Revisions #1 for details.

We now compared the accuracy and speed of DAS to that of other tools for mice (USVSEG), Bengalese and Zebra finches (TweetyNet), and marmosets (Oikarinen et al., 2019). DAS is as good as or better then these specialized methods, and between 3x and 20x faster. The comparisons are shown in a new Table 2 (reproduced on page 5).

We have also tested DAS on two more species – marmosets and Zebra finches. For both species, DAS performs excellently (F1 scores 94% for marmosets and 100% for Zebra finches). The two new figures and the two new sections from Results are reproduced on pages 2-4.

Our wording is now more accurate throughout, to reflect the outcome of these comparisons. For instance, when discussing our results on bird song (p7, l229):

“Thus, DAS performs as well as specialized deep learning-based methods for annotating bird song (Cohen et al., 2020a; Koumura and Okanoya, 2016) (Table 2).”

The methods should state very clearly where the data sets used are available for download. While the authors may believe in DeepSS, other challengers will inevitably arise. These authors should be given the opportunity to report head-to-head performance against DeepSS on the same data.

We do hope that others improve on our method! To aid direct performance comparisons, we deposited all data (audio and annotations) used for training and testing DAS here: https://data.goettingen-research-online.de/dataverse/das. In addition, we added a new Table 5 listing (1) all original data sources, (2) direct links to the data used by DAS for each species tested, and (3) links to the trained models including sample data and code for running the models (https://github.com/janclemenslab/das-menagerie).

The text references in the mouse section are to figure 3, but the relevant figure is now figure 2.

Thank you! We fixed all figure references.

There are several places where the color scale used is useless, namely in the confusion matrices (Figure 1D, 1I, 2E, 2H, 3C and some supplemental figs). At these performance levels, color just indicates the correct vs. incorrect parts of the graph – relative shades of the heat map are not perceptible.

We agree that the confusion matrices as shown are not very informative given the performance levels. We still prefer to use color instead of plain text tables to make the information indicated in the text labels of the confusion matrices more glanceable.

The birdsong confusion matrices are similarly uninformative, since all that is seen is a bunch of boxes along the diagonal. Worse yet, the discretization of the figure makes it appear that boxes along the diagonal have different sizes/offsets, suggesting structure where there is none. Showing marginals of the matrix like Figure S4E can be useful.

We now log-scaled the color code for the confusion matrix in Figure 3C and Figure 3 —figure supplement 1D (was S4D) to make annotations errors apparent. We also fixed the issue with the discretization of the figure.

In Figure S4C, precision and recall are plotted whereas in S4E false positive and false negatives are plotted. Are both plots needed?

Given that false positive and false negative rates are now more visible from the log-scaled confusion matrices in Figure 3 —figure supplement 1D, we removed panel S4E.